



# Revealing the dynamics of a local Alpine windstorm using large-eddy simulations

Nicolai Krieger[1], Heini Wernli[1], Michael Sprenger[1], and Christian Kühnlein[2]

[1]Institute for Atmospheric and Climate Science, ETH Zurich, Zurich, Switzerland
[2]European Centre for Medium-Range Weather Forecasts (ECMWF), Bonn, Germany

**Correspondence:** Nicolai Krieger (nicolai.krieger@env.ethz.ch)

**Abstract.**

The local atmospheric flow in mountainous terrain can be highly complex and deviate considerably from the ambient conditions. One example is a notorious local windstorm in a narrow and deep valley in north-eastern Switzerland, known as the Laseyer, that had previously even caused a train derailment. This windstorm is characterized by strong south-easterly winds blowing perpendicular to the valley axis during strong north-westerly ambient flow conditions. We investigate the mechanism of this local windstorm and its sensitivity to changes in the prescribed ambient wind using large-eddy simulation (LES). The LES are performed using the Portable Model for Multi-Scale Atmospheric Prediction (PMAP) at a horizontal grid spacing of $30\,\mathrm{m}$ and applying a terrain-following vertical coordinate with steep slopes of the real topography reaching nearly 80°. The simulations, driven by strong north-westerly ambient winds, successfully capture the flow reversal in the valley with quasi-periodically occurring short episodes of wind bursts regularly exceeding $20\,\mathrm{m\,s^{-1}}$ and in exceptional cases exceeding $35\,\mathrm{m\,s^{-1}}$. The flow reversal is explained by an amplifying interplay of (1) a recirculation region formed by flow separation in the lee of the upstream ridge, and (2) a vortex caused by a positive pressure anomaly formed by the north-westerly winds impinging on the downstream mountain. This formation mechanism is supported by a simulation in which the height of the downstream mountain is reduced, resulting in a decrease in the strength of the reversed in-valley flow. In agreement with previous observational studies, a series of simulations with modified ambient wind conditions reveal that the intense gusts ($> 20\,\mathrm{m\,s^{-1}}$) only occur in a narrow window of ambient wind directions and if its speed is at least $16\,\mathrm{m\,s^{-1}}$. Smoothing the topography in the LES reduces the maximum wind speeds in the target region by 10–30%. Overall, our semi-idealized LES in complex and steep terrain reveal the three-dimensional structure and the mechanism of the local windstorm. Moreover, they point to the importance of the local topography and its complex interplay with the three-dimensional and transient flow leading to the in-valley flow reversal and strong winds that characterize the Laseyer. The study further highlights the importance of the topographic details for the quantitatively correct simulation of atmospheric flows in complex terrain.

## 1 Introduction

Mountainous terrain modifies the large-scale atmospheric flow when air is forced to flow either over or around mountains (Jackson et al., 2013). Typical terrain-forced flow phenomena are downslope windstorms, mountain waves and lee waves



further downstream, gap winds and barrier jets, turbulent eddies or wakes, flow separation in the lee of a barrier, or the wind speed increase at the crest of a mountain (Whiteman, 2000; Jackson et al., 2013). Some of these phenomena can be severe, hazardous to aviation, or cause damage to infrastructure or mountain forests, also by enhancing wildfires (Whiteman, 2000). Many mountain effects are related to flow separation, which can occur when air is forced to flow over steep mountains (Stull, 1988). Under conditions of high wind speeds and weak stratification, the flow can separate in the lee of the mountain, creating

a turbulent wake where the air is mixed (Baines, 2022). This flow separation is often associated with a recirculation region and reversed flow at the surface compared to further aloft, and can occur on either sides of a barrier (Scorer, 1955; Whiteman, 2000). Physically, the flow separates as it encounters an adverse pressure gradient where it is decelerated until it separates from the surface (Ambaum and Marshall, 2005). If and where this separation occurs depends on several factors and will be discussed in more detail in the following. Note that the flow can also follow the topography and only separate further downstream, known

as post-wave separation (Baines, 2022). This flow separation is frequently associated with downslope windstorms and rotors further downstream, as extensively investigated within the Terrain-Induced Rotor Experiment (T-REX) project (Grubišić et al., 2008; Kühnlein et al., 2013; Strauss et al., 2016). While both the separation in the lee and further downstream share many aspects, such as recirculation, we will only focus on lee-side separation in the following.

  Baines (2022), using laboratory experiments, and Ambaum and Marshall (2005), using linear theory, predicted the separation

for turbulent flow over a two-dimensional hill to depend on two non-dimensional parameters. These are the aspect ratio $h/L$ of the obstacle, where $h$ is its height and $L$ its (downstream) half-width, and the non-dimensional mountain height $Nh/U$, where $U$ is the wind speed and $N$ is the Brunt-Väisälä or buoyancy frequency (see, e.g., Figs. 4 and 5 in Ambaum and Marshall, 2005). For small values of the parameter $Nh/U$ and small aspect ratios, no separation occurs. For larger aspect ratios, the flow separates on the lee side of the hill. This behavior can be explained by the fact that for small values of $Nh/U$, the

natural wavelength of the flow is longer than the width of the hill (Stull, 1988). For large values of the parameter $Nh/U$, separation does not occur at the lee of the hill but may still occur further downstream. Following Baines (2022), this is called post-wave separation and is often associated with lee waves or rotors. Hunt and Snyder (1980) and Baines (2022) described this transition from lee-side to post-wave separation and the details of the recirculation region in more detail. At neutral stratification ($N = 0\,\mathrm{s}^{-1}$), the flow separates at the top of the obstacle or even upstream of it. With higher stability of the

approaching turbulent flow, the separation point moves downstream and the extent of the recirculation region becomes smaller. This is consistent with linear theory, which predicts the separated region to be compressed (Baines, 2022). Similar results for the role of the aspect ratio for the occurrence of flow separation have also been found in studies of wind tunnel experiments (Arya et al., 1987) or idealized numerical simulations (Wood, 1995; Kim et al., 2001).

  In addition to these two parameters, the shape of the obstacle and the surface characteristics also decisively determine the

occurrence of flow separation and the characteristics of the recirculation region. Linear theory by Ambaum and Marshall (2005) and Finnigan and Belcher (2004) predicted that friction facilitates flow separation. Similar results have also been obtained using numerical simulations (Wood, 1995; Sogachev et al., 2004), where a rougher surface (e.g., forest canopy) led to flow separation at lower aspect ratios than a smoother surface in a neutrally stratified flow. Furthermore, observations indicated that flow separation often occurs at salient edges, which not only facilitate flow separation but also determine the location of



separation (Scorer, 1955). In addition, surface heating also affects flow separation. Ambaum and Marshall (2005) suggested that diabatic heating suppresses flow separation, which has been confirmed in numerical simulations (Dörnbrack and Schumann, 1993; Allen and Brown, 2006). In contrast, Scorer (1955) observed that cooling of the air and the associated katabatic winds prevent flow separation whereas heating facilitates it. This agrees with the results by Mason (1987): strong heating increases the size and strength of the separation region. Lewis et al. (2008b) were able to resolve this contradiction with simulations,

which demonstrated that the effect of surface heating depends on the aspect ratio. For large aspect ratios (steep slopes), surface heating enhances flow separation and both the size and strength of the recirculation are increased (Lewis et al., 2008b). For flow over obstacles with small aspect ratio, however, surface heating can suppress flow separation (Lewis et al., 2008b). Further observations suggested that the interaction is even more complicated and that the occurrence of recirculation regions also depends on other factors, such as the interaction with thermally-driven winds (Lewis et al., 2008a).

Most of the aforementioned studies considered flow separation behind an obstacle. However, for sufficiently steep obstacles separation can also occur upstream of it (Scorer, 1955; Baines, 2022). This was illustrated in an idealized setting for an obstacle with a bluff forward face (Hunt et al., 1978), but was also reported for an obstacle with a slope greater than 60° (Tsoar, 1983), and in front of a cliff where a helical flow pattern was observed (Tsoar and Blumberg, 1991).

In recent years, several studies have investigated the occurrence and characteristics of mountain-induced recirculations at

specific locations using measurements and simulations. Simulations by Ma and Liu (2017) explored the flow over Bolund Hill, Denmark, focusing on the recirculation regions both upstream and downstream of the hill. At the Perdigão site in Portugal, lidar measurements revealed frequent flow recirculation between two parallel ridges when the wind is perpendicular to the ridge lines (Menke et al., 2019). In particular, recirculation is favored by high wind speeds and suppressed under stable atmospheric conditions (Menke et al., 2019), confirming earlier studies in idealized settings. Similarly, Lehner et al. (2019) identified high

wind speeds as a key factor favoring recirculation in the lee of a crater rim. In the Swiss Alps, recirculation was found behind a mountain peak, affecting the distribution of snow (Gerber et al., 2017). Furthermore, Menke et al. (2019) reported increased turbulence intensity at the downwind ridge during periods of recirculation. Strong turbulence behind a steep peak was also associated with flow separation in Strauss et al. (2015).

Many of the descriptions of flow behind hills or around obstacles, and particularly the turbulent wake region, are typically

presented as mean flow patterns (e.g., Hunt et al., 1978; Hunt and Snyder, 1980; Ma and Liu, 2017; Lehner et al., 2019; Menke et al., 2019). However, there is considerable unsteadiness and variability, and the wake does not form a closed recirculation bubble, but the turbulent region extends downstream (Hunt and Snyder, 1980; Kühnlein et al., 2013; Baines, 2022). Even in a two-dimensional representation, most separated regions are open, meaning that flow enters and leaves the region of flow separation (Belcher and Hunt, 1998). Furthermore, the flow field around an obstacle strongly fluctuates and substantial lateral

oscillations and asymmetries occur (Hunt and Snyder, 1980). Despite these attempts to describe the recirculation region beyond portraying the mean flow field, these studies did not quantify extreme gusts in the turbulent recirculation region and did not relate them to potential damage. However, according to Grønås and Sandvik (1999), mountain lee side separation is believed to be the main mechanism behind major wind damage in Norway. Specifically, they reported that power pylons in a short, narrow valley in northern Norway broke down during the passage of a severe extratropical cyclone. Interestingly, the power pylons fell



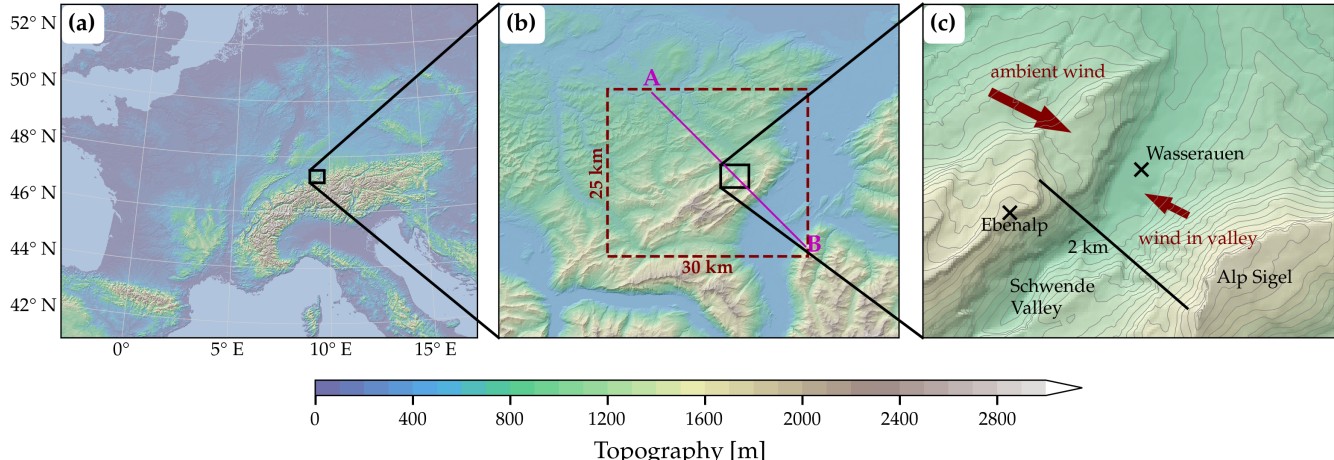

**Figure 1.** Topography of **(a)** western and central Europe, **(b)** north-eastern Switzerland, and **(c)** the target region with the Schwende valley and its surroundings. In **(b)**, the domain used for the simulations is indicated by the dashed red rectangle, and in **(c)**, the main wind directions during Laseyer conditions and a distance of 2 km across the valley are indicated. The gray lines in **(c)** show the height of the topography (interval 50 m).

opposite to the ambient wind direction and the damage was probably caused by wind gusts exceeding $50\,\mathrm{m\,s^{-1}}$ (Grønås and Sandvik, 1999).

Damage due to local winds blowing opposite to the ambient wind direction was also reported in a narrow valley in the Alpstein massif in north-eastern Switzerland (Fig. 1). There, the local windstorm — called Laseyer — caused a train to derail in the narrow Schwende valley (Fig. 1c) during the winter storm Kyrill in January 2007 (see Fink et al., 2009, for a more

general description of the storm and its impacts). This derailment was likely caused by very strong south-easterly winds in the narrow valley during north-westerly ambient wind conditions (Sprenger et al., 2018, and see Fig. 1c). Today, warnings are issued and the service of the local train is suspended when the local windstorm and associated strong gusts are likely to occur (Sprenger et al., 2024).

A 10-year climatology of Laseyer observations is presented in Egloff (2009) and Sprenger et al. (2018) and summarized

in the following. The strongest gusts with an easterly component in the valley occur almost exclusively when the winds at the measurement station Ebenalp on the upstream ridge (see Fig. 1c) blow from westerly to north-westerly directions (265° to 295°). Moreover, the strongest westerly to north-westerly winds measured on the upstream ridge (Ebenalp) are associated with easterly winds in the valley (Wasserauen). Analyzing the temporal evolution of the atmospheric flow before, during, and after a Laseyer event, Sprenger et al. (2018) found that with the onset of the Laseyer, the wind direction on the upstream ridge

changes from westerly to more north-westerly. Furthermore, the main season of the Laseyer is during the cold season from about October to March. This is consistent with the requirement for strong westerly to north-westerly ambient winds for the occurrence of the Laseyer, which coincides typically with the passage of winter storms.





Although the Laseyer and its climatology are well known, the specific mechanism leading to the very strong gusts near the valley floor is not yet understood. Sprenger et al. (2018) listed several potential mechanisms that could lead to the flow reversal and the strong gusts, but they could not identify the causing mechanism. Flow separation and recirculation bubbles, however, were speculated to be of great importance. The objective of this high-resolution modeling study is to identify the mechanism that leads to the flow reversal and strong gusts through dry semi-idealized large-eddy simulations (LES) over the Alpstein massif using the new Portable Model for Multi-Scale Atmospheric Prediction (PMAP; see Section 2.1), which builds on previous developments at ECMWF (Smolarkiewicz et al., 2014, 2016; Kühnlein et al., 2019). The study aims to investigate the mechanism of the Laseyer and analyze the sensitivity of the flow in the narrow valley to changes in the ambient flow conditions.

The paper is organized as follows. In Section 2, we introduce the model and the model setup; in Section 3, a simulation exhibiting flow reversal in the narrow valley is presented and the mechanism of the flow reversal and high wind speeds is discussed. In Section 4, the sensitivity of the flow in the narrow valley to changes in the ambient flow conditions is investigated; and the main conclusions are drawn in Section 5.

## 2 Model description and simulation setup

### 2.1 Numerical model

The present study is based on the LES configuration of the Portable Model for Multi-Scale Atmospheric Prediction (PMAP; Kühnlein et al.). PMAP is implemented in a productive Python programming environment and provides high-performance execution across computing architectures including systems with accelerators [1]. The numerical model solves the fully compressible equations using semi-implicit integration that is combined with conservative flux-form advective transport. The semi-implicit compressible integration provides unconditional stability with respect to fast acoustic and buoyant modes in all spatial dimensions and builds on the design developed in Smolarkiewicz et al. (2014). Even though various advection schemes are available in PMAP, all results in this paper were produced exclusively using the multidimensional positive-definite advection transport algorithm (MPDATA) in its infinite-gauge version with non-oscillatory enhancement (Smolarkiewicz and Grabowski, 1990; Smolarkiewicz and Margolin, 1998; Waruszewski et al., 2018). Subgrid-scale turbulent transport is implemented by means of a three-dimensional flux-form diffusion scheme with vertically-implicit horizontally-explicit time stepping and where exchange coefficients are modeled using a prognostic 1.5-order turbulence kinetic energy (TKE) closure following Deardorff (1980) and Stevens et al. (1999). All aspects of the model are rigorously implemented in curvilinear coordinates to accommodate general terrain-following vertical model levels (Kühnlein et al.). Implementation of boundary conditions in the semi-implicit scheme of the model adopt the approach explained in Smolarkiewicz et al. (2007). Both moist processes and radiation effects in the model are neglected for the present study. Furthermore, the bulk formula of the neutral drag law based on Monin-Obukhov similarity theory is employed to calculate the drag coefficient and associated surface fluxes of momentum, while surface heat

---

[1]Hardware portability and high performance of PMAP is achieved by implementation with the GridTools for Python (GT4Py; https://github.com/GridTools/gt4py) domain-specific software framework (Afanasyev et al., 2021); see also Ben-Nun et al. (2022) and Ubbiali et al. (2024) for recent applications.





fluxes are assumed to be zero. The incorporation of the surface momentum fluxes in the turbulence scheme completely accounts
for sloped surfaces as described in Epifanio (2007) and thus avoids the flat-boundary assumption.

## 2.2    Simulation setup

The simulation setup considers real surface fields and topography, while the initial and boundary conditions are idealized. The simulations are performed on an $f$-plane located at 45° N, over a domain of $30\,\mathrm{km} \times 25\,\mathrm{km}$, and up to a height of $20\,\mathrm{km}$. The lateral boundaries are assumed to be open, while a free-slip rigid lid is implemented at the upper boundary. The
horizontal grid spacing is set at a constant value of $\Delta = \Delta x = \Delta y = 30\,\mathrm{m}$, and the vertical grid spacing is $\Delta z = 15\,\mathrm{m}$ close to the surface and stretched to $\Delta z \approx 375\,\mathrm{m}$ at the domain top. The lowest model level is located $7.5\,\mathrm{m}$ above the ground. This resolution corresponds to $1001 \times 835 \times 151$ grid points in the $x$-, $y$-, and $z$-directions, respectively. For the distribution of the terrain-following vertical levels, we use an algorithm following Klemp (2011), with the differences outlined in Appendix A1.

Over the uppermost $6\,\mathrm{km}$ of the model domain, a damping layer is applied with a time scale of $\tau = 300\,\mathrm{s}$. At the lateral
boundaries, absorbers with a time scale of $\tau = 30\,\mathrm{s}$ are applied over the outer $1500\,\mathrm{m}$ of the domain. Within these absorption layers, the three velocity components are relaxed towards the inflow profile. The simulations use time steps $\Delta t$ of $\frac{8}{64}\,\mathrm{s}$, $\frac{8}{64}\,\mathrm{s}$, $\frac{5}{64}\,\mathrm{s}$, and $\frac{4}{64}\,\mathrm{s}$ for the simulations with an ambient geostrophic wind speed $U_g$ of $12\,\mathrm{m\,s^{-1}}$, $16\,\mathrm{m\,s^{-1}}$, $20\,\mathrm{m\,s^{-1}}$, and $24\,\mathrm{m\,s^{-1}}$, respectively. All simulations run for $3\,\mathrm{h}$, with the first hour considered spin-up and the subsequent $2\,\mathrm{h}$ of simulation used for analysis. During these $2\,\mathrm{h}$ of simulation, output is written to file every $10\,\mathrm{s}$. The temporal average over the period from 1 to $3\,\mathrm{h}$
simulation time is referred to as the *simulation mean*, or *mean wind* when referring to the wind.

### 2.2.1    Initial and lateral boundary conditions

The simulations are forced by a uniform ambient wind blowing from a westerly to north-westerly direction. This wind is prescribed over the entire depth of the domain and remains constant throughout the simulation. It is in geostrophic balance with a horizontally constant ambient pressure gradient. To investigate the sensitivity of the flow in the target region, the ambient wind
speed $U_g$ is varied between 12 and $24\,\mathrm{m\,s^{-1}}$, and the wind direction between 285° and 330°. The simulations are conducted using a uniformly stratified atmosphere with a sea level pressure of $1015\,\mathrm{hPa}$, a temperature of $280\,\mathrm{K}$ at sea level, and a Brunt-Väisälä frequency of $N = 10^{-2}\,\mathrm{s^{-1}}$. The two simulations used to investigate the role of stratification (Sect. 4.3) are forced with an ambient geostrophic wind with $U_g = 20\,\mathrm{m\,s^{-1}}$ from 300°, and the Brunt-Väisälä frequency is changed to $N = 1.5 \cdot 10^{-2}\,\mathrm{s^{-1}}$ and $N = 7 \cdot 10^{-3}\,\mathrm{s^{-1}}$, respectively. The initial condition for the total wind field is found from the ambient wind by projection
on the anelastic mass continuity constraint, which requires the iterative solution of a three-dimensional elliptic boundary value problem (Smolarkiewicz and Margolin, 1994).

To spin up the turbulence in our simulations, random perturbations are applied to the potential temperature field at initialization. These perturbations decrease linearly in amplitude with height up to $500\,\mathrm{m}$ above the topography. We use the cell perturbation method described in Muñoz-Esparza et al. (2015) with an amplitude of $\tilde{\theta}_p = U_g^2/(c_p \mathrm{Ec})$ at the surface, but with
only one perturbation in the vertical spanning the entire $500\,\mathrm{m}$ of the perturbation region. Here, $\mathrm{Ec} = 0.18$ is the Eckert number





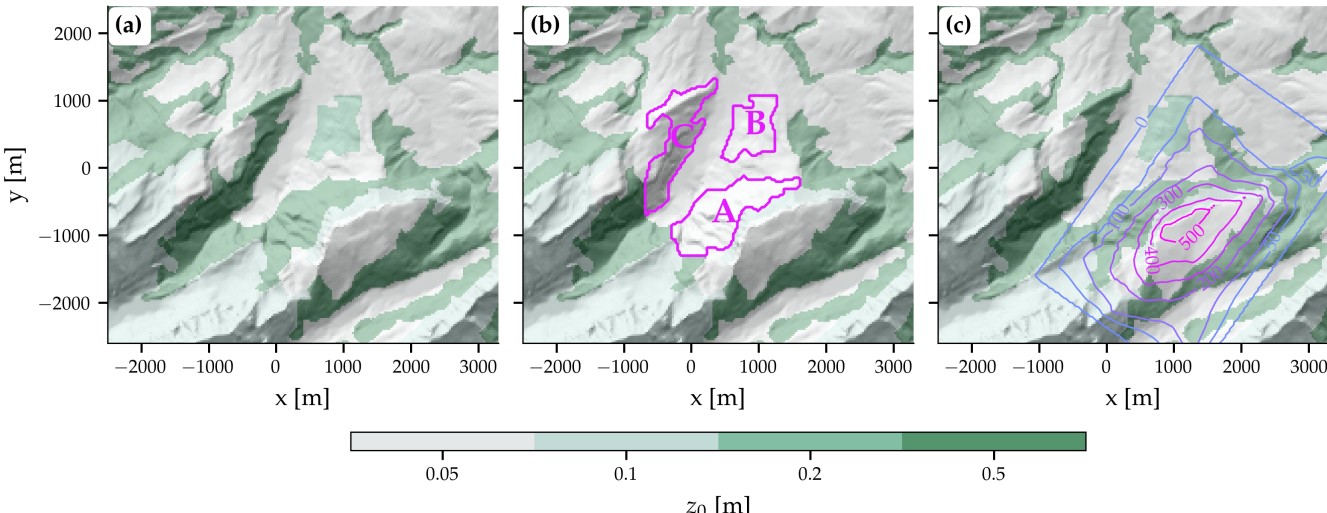

**Figure 2.** Topography (hill shading) and the roughness length field $z_0$ (color shaded) in the target region (see Fig. 1c). Shown are **(a)** the realistic fields used in most simulations, **(b)** the fields used for one sensitivity experiment where certain areas (labeled A, B, C) are replaced by grass, and **(c)** the modified topography used in another sensitivity experiment with the difference to the real topography highlighted (colored lines, in m).

and $c_p$ the specific heat capacity at constant pressure. Furthermore, these perturbations are applied using a cell size of eight grid points in both horizontal directions.

During the simulations, random perturbations with the same perturbation amplitude as the initial perturbations are applied to the north and west boundaries with a width of one perturbation cell or eight grid points (and again the same perturbation for all perturbed vertical levels). The perturbations are applied using a perturbation time scale $t_p$ following Muñoz-Esparza et al. (2015). The value of the perturbation time scale is calculated as $t_p = d_c/U_g$, where $d_c = 8\sqrt{2}\Delta$ is the diagonal of a perturbation cell (Muñoz-Esparza et al., 2015).

#### 2.2.2 Lower boundary condition

At the lower boundary, we prescribe a spatially variable roughness length $z_0$ based on land cover data. To achieve this, the 2018 CORINE land cover dataset (European Environment Agency, 2023) is used, which has a horizontal resolution of $100\,\mathrm{m}$. The CORINE land-use categories are translated into constant $z_0$ values using the winter values in the lookup table by Pineda et al. (2004). We select the land cover category at the center of the grid point to specify the roughness length (Fig. 2a). For the two sensitivity experiments presented in Sect. 3.4, the roughness length for grass is used instead of the roughness length for forest in the lower part of the valley. In the first simulation, areas A and B in Fig. 2b are replaced, and in the second simulation, area C in Fig. 2b is replaced.



To specify the height of the topography, we use the matrix model of the DHM25 dataset[2] provided by the Federal Office of Topography (swisstopo) at a horizontal resolution of $25\,\mathrm{m}$. For the reference simulation presented in Sect. 3, we do not apply any smoothing of the topography and just interpolate the topography to the grid of the simulation (Fig. 2a). The use of the high-resolution topography dataset results in a maximum slope of $77.4°$ in the discrete representation of the simulation. For the set of sensitivity simulations presented in Sect. 4, we smooth the topography to remove scales smaller than about $6\Delta$ with a high-order Shapiro filter (Shapiro, 1975). Smoothing the topography is customary in numerical models to remove forcing at the grid scale and, for simulations in complex terrain, reduce the maximum slopes within the simulation domain. The smoothing procedure, which is explained in Appendix A2, leads to a reduction of the steepest slopes to $70.5°$. Smoothing was only applied to the simulation presented in Sect. 4 as these simulations were performed before we realized the excellent quality of the simulation without any topography smoothing presented in Sect. 3.

At the domain boundaries, we set the topography to a constant value, to ensure that uniform boundary conditions can be easily prescribed and the mass in the model domain remains constant over time. A smooth transition from the constant height of the topography at the lateral boundaries to the real topography in the domain interior is performed using a cosine function, as described in detail in Appendix A3.

For a simulation conducted to investigate the role of the downstream mountain Alp Sigel (described in Sect. 3.5), we modified the topography in the target region as illustrated in Fig. 2c. A description of how we applied these topographic changes is given in Appendix A4.

## 3 Understanding the mechanism of the local windstorm

We begin our analyses by presenting the simulation with a prescribed ambient geostrophic wind of $U_g = 20\,\mathrm{m\,s^{-1}}$ from $300°$. This simulation reproduces the observed flow reversal in the narrow Schwende valley under strong north-westerly ambient winds, resulting in very high wind speeds at the valley floor. It provides an ideal case for investigating the local flow dynamics and identifying the cause of the strong south-easterly gusts in the valley. Hereafter, we will refer to this simulation as the reference simulation.

### 3.1 General flow description

Following a spin-up period of one hour, the simulation reveals a well-developed turbulent boundary layer in the domain's interior (Fig. 3). The depth of the boundary layer is between $500\,\mathrm{m}$ and $1000\,\mathrm{m}$ over most of the domain, while over the wide Rhine Valley on the south-eastern boundary of the domain, the boundary layer is significantly deeper, extending to a depth of approximately $1500\,\mathrm{m}$ (Fig. 3). Near the western and northern boundaries, the complex topography and random perturbations of potential temperature (see Sect. 2.2.1) cause turbulence to develop within a few kilometers of the domain boundaries (near A in Fig. 3).

---

[2]https://www.swisstopo.admin.ch/en/geodata/height/dhm25.html



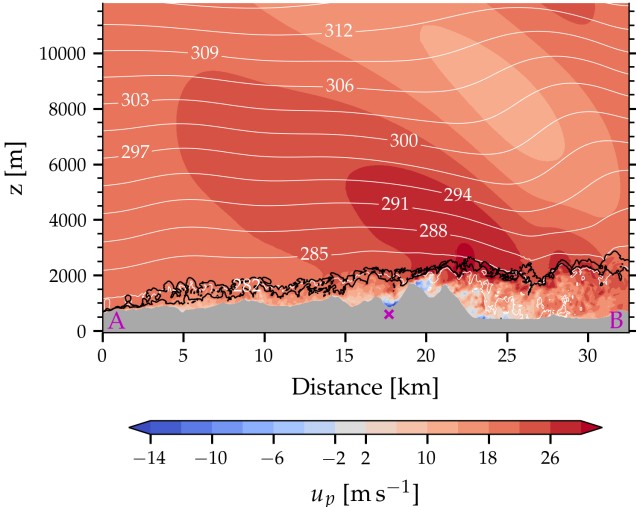

**Figure 3.** Vertical section across the model domain and valley (location indicated by the magenta line in Fig. 1b) of the flow field after 1 h of simulation time in the reference simulation. Shown are the horizontal wind speed parallel to the cross section ($u_p$, color shaded, north-westerly winds are positive, south-easterly winds negative), isentropes (white lines, in K), and subgrid-scale TKE (black lines for $0.001\,\mathrm{m^2\,s^{-2}}$ and $0.1\,\mathrm{m^2\,s^{-2}}$). The magenta cross in the middle of the cross-section marks the Schwende valley.

On the lowest few model levels, the wind field exhibits large variations. Over the more hilly terrain to the north-west of the Alpstein massif, the winds are predominantly westerly to north-westerly (not shown). The winds are channelled along the Alpstein massif and become more south-westerly over the northern slopes of the Alpstein massif. Over the Rhine Valley at the south-eastern boundary of the domain and in the very complex topography in the interior of the Alpstein massif, almost any
wind direction occurs (not shown).

A mountain gravity wave is triggered by the Alpstein massif as a whole, resulting in wind speeds increasing up to $8\,\mathrm{m\,s^{-1}}$ in the free atmosphere over the Alpstein massif just above the turbulent boundary layer (Fig. 3). The individual ridges and mountain peaks of the Alpstein massif do not induce mountain waves, as would be expected from their short horizontal scale, the moderate vertical stability of the atmosphere, and the high horizontal wind speeds (see Durran, 1990). As a result, wind
speeds are generally highest at the mountain tops and more uniform and — except for the perturbations due to the triggered mountain wave — more similar to the imposed ambient wind speed at higher altitudes (Fig. 3).

## 3.2 Flow patterns in the valley and its surroundings

Figure 4 displays the evolution of the surface wind in the valley and its immediate surroundings during a short period of the simulation. An animation covering a longer period of the simulation is available in the supplementary material. Strong
westerly to north-westerly winds prevail at exposed locations such as the ridge to the north-west and the ridge and mountain to the east and south-east of the valley (Alp Sigel). The winds on the steep slopes north-west of the valley are generally light.





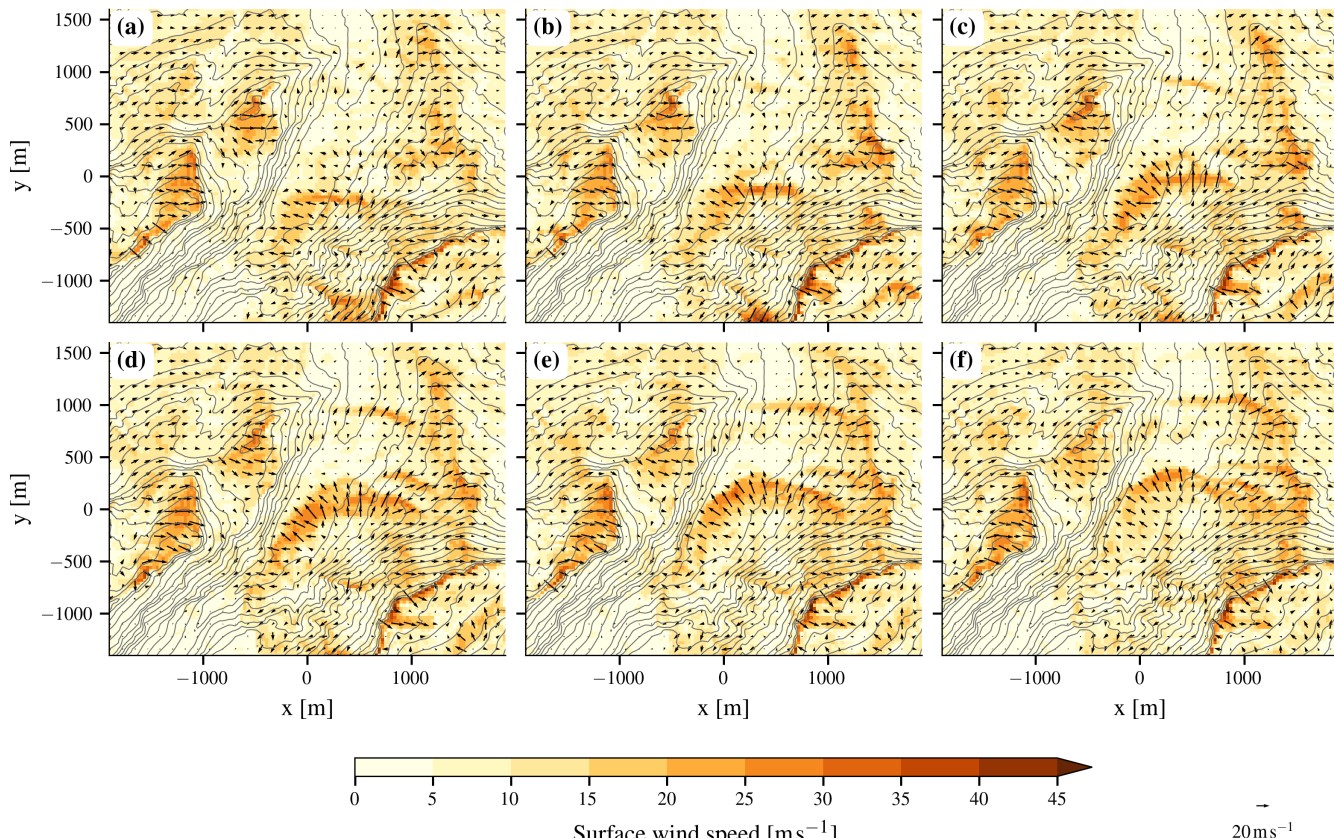

**Figure 4.** Flow evolution in the Schwende valley and its surroundings over 50 s of simulation time. Shown are surface wind (arrows, color shaded for wind speed) and the height of the topography (gray lines, interval 50 m) at **(a)** 6440 s, **(b)** 6450 s, **(c)** 6460 s, **(d)** 6470 s, **(e)** 6480 s, and **(f)** 6490 s simulation time in the reference simulation.

Over the ridge north-west of the valley, the flow separates from the surface as it encounters the very steep slopes. The temporal evolution of the surface wind in the target region indicates that the flow in the valley is highly turbulent. Strong easterly to south-easterly winds are embedded in this turbulent flow at the valley floor (e.g., Fig. 4d,e). These are frequently associated
with momentary stagnation points on the escarpment to the south-east of the valley and rings or segments of higher wind speeds that spread out radially. This key result is evident from Fig. 4 and can be best seen in the animation with 10 s time resolution in the supplementary material. The stagnation points and highest wind speeds vary in location during the simulation. Between periods of high wind speeds, some parts of the valley floor experience calm conditions (e.g., Fig. 4a).

We continue our analysis by examining some basic statistical properties of the surface wind in the target region. The mean
surface wind field (Fig. 5a) reveals strong westerly to north-westerly winds at exposed locations, such as the ridge upstream of the valley, which is consistent with the patterns observed in the instantaneous flow fields. At the valley floor, easterly to south-easterly winds prevail, which reach average wind speeds of about 12 m s$^{-1}$. These mean south-easterly winds are significantly





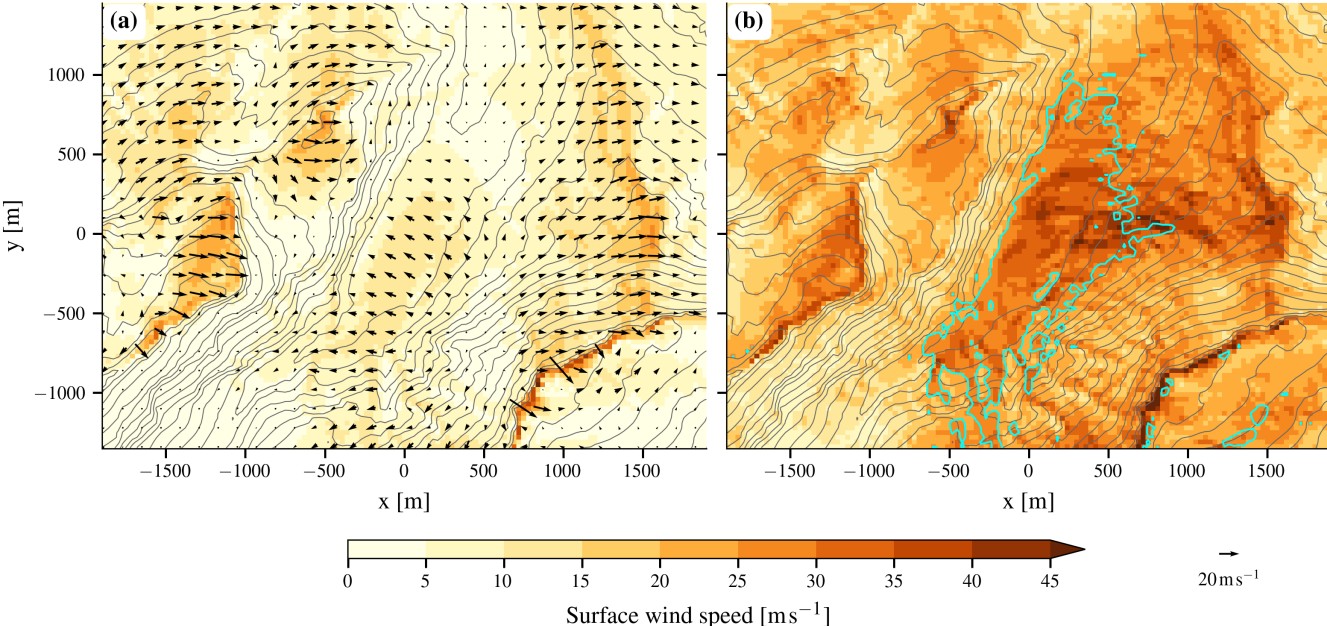

**Figure 5.** Surface wind in the target region. Shown are **(a)** mean surface winds (arrows, color shaded for wind speed), averaged from 1 h to 3 h of simulation time, and **(b)** maximum surface wind speed during the same period. The gray lines depict the height of the topography (interval is 50 m), while the cyan lines in **(b)** identify the regions where easterly to south-easterly ($65° \leq \varphi \leq 165°$) winds with a speed of at least $25\,\mathrm{m\,s^{-1}}$ occur.

weaker than the maximum wind speeds (Fig. 5b) that occur during the simulation. They are based on discrete output times and therefore represent a lower limit of the maximum wind speeds, indicating that likely higher wind speeds occurred during the simulation. The maximum wind speeds at the valley floor exceed $30\,\mathrm{m\,s^{-1}}$ near the measurement station of Wasserauen and $40\,\mathrm{m\,s^{-1}}$ at certain locations on the valley floor (Fig. 5b). These very high wind speeds have an easterly to south-easterly direction, and they are comparable in magnitude to those occurring on the upstream ridge. The highest wind speeds in the valley occur in our LES near the ground, i.e. on one of the three lowest model levels. The area where mean wind speeds of at least $10\,\mathrm{m\,s^{-1}}$ occur is relatively small, measuring about 1500 m along the valley axis and 500 m in the cross-valley direction (Fig. 5a). The region where maximum wind speeds of at least $25\,\mathrm{m\,s^{-1}}$ from an easterly to south-easterly direction occur is larger and includes the lowermost part of the valley where the mean wind speed is low (Fig. 5b).

A simulated time series of the surface wind for the grid point closest to the measurement station Wasserauen (Fig. 6) reveals the turbulent nature of the flow in the valley, including instances when the surface wind speed exceeds $30\,\mathrm{m\,s^{-1}}$. Although the wind direction is mostly easterly to southerly, it varies significantly during the simulation. At certain moments, there are weak winds with a westerly component (e.g., at 6300 s or 9700 s simulation time). Periods of high wind speeds ($> 25\,\mathrm{m\,s^{-1}}$) are of very short duration (typically 10 to 30 s) and before and after, the wind speed is considerably weaker. Contrary, periods of weak wind speeds ($< 6\,\mathrm{m\,s^{-1}}$) are usually of longer duration (e.g., at 9700 s or 6400 s simulation time). This is consistent with the



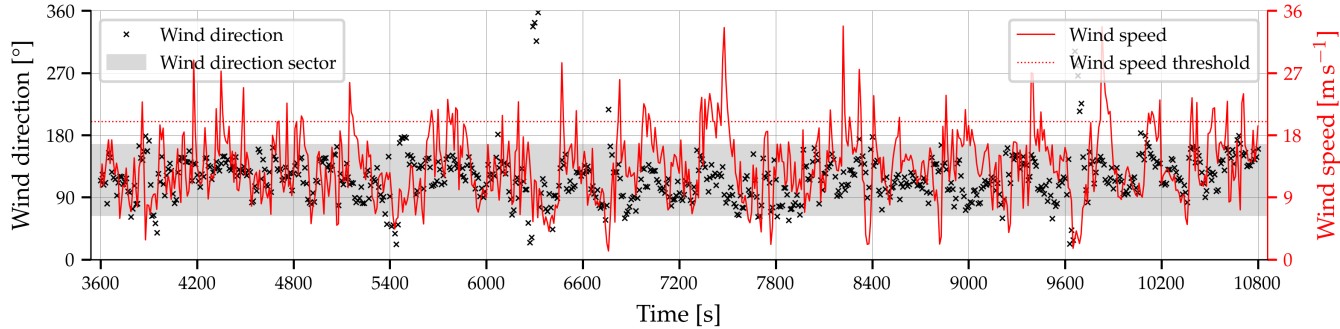

**Figure 6.** Simulated time series of surface wind at the location of the measurement station Wasserauen on the valley floor. The wind direction sector and wind speed threshold refer to the criteria used for the gust composites (Sect. 3.3). The temporal resolution is $10\,\mathrm{s}$.

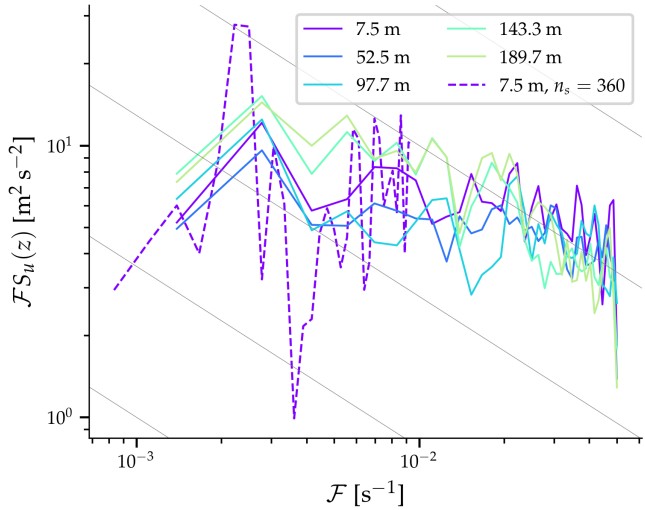

**Figure 7.** Spectra of the streamwise velocity component at different heights above the valley floor. The dashed line represents the power spectral density of the streamwise component of the surface wind ($z = 7.5\,\mathrm{m}$) using a longer segment length of $n_s = 360$. Note that the $y$-axis represents the product of frequency and the power spectral density. The thin gray lines indicate the $-2/3$ spectral slope.

distribution of wind speeds, which is weakly positively skewed (skewness of 0.5, not shown). However, there is a prolonged period of wind speeds exceeding $20\,\mathrm{m\,s^{-1}}$ from approximately $7300\,\mathrm{s}$ to $7500\,\mathrm{s}$ of simulation time (Fig. 6).

Furthermore, the time series indicates that high wind speeds are distributed relatively evenly throughout the simulation (Fig. 6). This distribution may suggest a quasi-periodic pattern in the highest wind speeds, such as pulsations. The animation of the surface wind in the supplementary material also qualitatively indicates such a quasi-periodic behavior of the flow in the narrow valley. To further investigate this, we conduct a spectral analysis of the time series of the streamwise component of the surface wind presented in Fig. 6. In addition to providing information about (quasi-)periodic behavior, a spectral analysis also reveals

information about the turbulence characteristics and how it is represented in the simulation (e.g., Wyngaard, 2010). The method



utilized for this analysis is Welch's averaged periodogram (Welch, 1967). It is a widely used method as it reduces the variability within the periodogram and, consequently, gives a more robust estimate of the spectral density function by breaking up the time series into overlapping segments, calculating their periodograms, and averaging them (Percival and Walden, 1993, p. 294). This method is applied to the time series between $1\,h$ and $3\,h$ simulation time using a segment length of $12\,min$ or $n_s = 72$

output time steps. Figure 7 displays the power spectral density $S_u$ of the streamwise wind component on model levels 1, 4, 7, 10, and 13, corresponding to approximately $7.5\,m$, $52.5\,m$, $97.7\,m$, $143.3\,m$, and $189.7\,m$ above ground, respectively. Firstly, following Högström et al. (2002), two different spectral regimes can be identified. At lower frequencies $\mathcal{F}$, and particularly close to the ground, the spectra of the streamwise velocity component indicate only a weak slope corresponding to the energy input range of turbulence. The $-2/3$ spectral slope corresponding to the inertial subrange (Wyngaard, 2010) is visible at higher

frequencies, particularly for spectra taken at higher elevations. Based on these spectra, we argue that PMAP represents the turbulent fields well, the inertial subrange exists in our simulations, and the use of the term LES is appropriate (Cuxart, 2015). Secondly, we identify the frequency of the fluctuations that contain the most energy. To allow for more variability within the periodogram, we have included the power spectral density calculated using a longer segment length ($n_s = 360$ output time steps, equivalent to $1\,h$) for a part of the frequency range (indicated by the dashed line in Fig. 7). The frequency with the highest

energy is $\mathcal{F} \approx 2.2 \cdot 10^{-3}\,s^{-1}$, corresponding to a period of $450\,s$. Fluctuations at slightly higher frequencies contain much less energy, while even higher frequencies contain again more energy (which then transitions into the inertial subrange). The local minimum of the power spectral density occurs at a frequency of $\mathcal{F} \approx 3.6 \cdot 10^{-3}\,s^{-1}$ or a period of about $280\,s$. Therefore, this spectral analysis confirms the quasi-periodic behavior of the surface winds in the narrow valley, and we regard this as an important finding of this study, which was not known from previous investigations of the Laseyer phenomenon. However, we

will not investigate the source of these pulsations any further, but focus on the flow associated with the strongest gusts next.

### 3.3  Gust composites

In the following, we investigate the flow in the narrow valley and particularly the highest wind speeds in more detail. To achieve this, we calculate composites at times of the highest wind speeds, referred to as Laseyer gusts, on the valley floor at the location Wasserauen (see Fig. 1c). A Laseyer gust is defined as a wind speed of at least $20\,m\,s^{-1}$ and an easterly to south-easterly wind

direction ($65° \leq \varphi \leq 165°$), as illustrated by the thresholds in Fig. 6. Therefore, for the composite, we choose only time steps when the wind has a significant component perpendicular to the valley axis and the flow at Wasserauen is reversed compared to the ambient flow aloft. The wind speed threshold is set to include approximately 10% of the output time steps in the composite.

Figure 8 presents horizontal sections through Laseyer gust composites in the target region, together with deviations from the simulation mean indicated by the colored lines. Gusts at the valley floor are associated with stronger winds on the valley

floor also a few hundred meters along the valley axis (Fig. 8a). Additionally, southerly to south-westerly winds on parts of the downstream (here and in the following regarding the ambient flow direction) escarpment are stronger during Laseyer gusts. Thus, a Laseyer gust is associated with a half circle of increased wind speed at the valley floor and the lowest part of the downstream escarpment. Laseyer gusts are also associated with a divergent surface wind field at the slopes of the downstream





**Figure 8.** Laseyer gust composites (see text for definition). Shown are **(a)** surface wind, **(b)** vertical wind at an altitude of 1050 m, **(c)** surface pressure anomaly, and **(d)** pressure anomaly at an altitude of 1250 m. The pressure anomalies in **(c,d)** are calculated relative to a geostrophic flow over flat terrain. Gray shading and gray lines (interval is 50 m) indicate the height of the topography and colored lines the deviation of the gust composite from the simulation mean. Note that in **(d)**, the colored lines depict the deviation of the wind speed (and not the pressure anomaly) in the gust composite from the simulation mean. The magenta lines in **(a)** indicate the location of the vertical cross-sections presented in Fig. 9.



mountain (Fig. 8a, near $(x, y) = (400\,\mathrm{m}, -300\,\mathrm{m})$). The surface wind field in the valley's surroundings exhibits no or only
minimal deviations from the simulation mean, illustrating the rather localized nature of the Laseyer gusts.

At an altitude of $1050\,\mathrm{m}$, i.e., approximately 200 m above the valley floor, strong upward motion reaching up to $10\,\mathrm{m\,s^{-1}}$
is simulated on the very steep slopes north-west of the valley, while downward motion of up to $8\,\mathrm{m\,s^{-1}}$ occurs on the south-
eastern side (Fig. 8b). During Laseyer gusts, these downward motions intensify over the lower part of the valley. Consistent
with these enhanced upward and downward motions compared to the simulation mean, westerly to north-westerly winds up to
$8\,\mathrm{m\,s^{-1}}$ stronger than the simulation mean are simulated during Laseyer gusts at a height of $1250\,\mathrm{m}$ (Fig. 8d).

To explain these Laseyer gusts, we present the pressure anomaly compared to an atmosphere in geostrophic balance over
flat terrain in Figs. 8c,d at the surface and at an altitude of $1250\,\mathrm{m}$, respectively. The pressure anomaly at the surface exhibits
negative values over the crests and at exposed locations. This can be explained by the conservation of the Bernoulli function
when the winds are accelerated over the crests (see e.g., Gill, 1982). Additionally, a strong positive pressure anomaly is
visible on the downstream slope of the valley. During Laseyer gusts, both the positive pressure anomaly over the downstream
escarpment and the negative pressure anomaly at the valley floor are intensified compared to the simulation mean, as illustrated
by the colored lines in Fig. 8c. In the gust composite, a pressure anomaly difference of approximately $3.5\,\mathrm{hPa}$ is simulated
over a distance of roughly $1\,\mathrm{km}$ across the valley axis. Nonhydrostatic pressure anomalies with a similar magnitude were also
found in simulations by Grønås and Sandvik (1999). The pressure anomaly at $1250\,\mathrm{m}$ (Fig. 8d) exhibits a similar pattern with
a positive anomaly over the downstream escarpment and a negative anomaly in the center of the valley. At this altitude, the
pressure anomaly difference of about $3\,\mathrm{hPa}$ between the centers of the positive and negative anomalies leads to a slightly
weaker pressure gradient.

Figure 9 presents vertical sections through the Laseyer gust composite along two different lines. Notably, high wind speeds
are simulated at higher elevations and over exposed locations on both the upstream and downstream mountains (Fig. 9a,c). In
contrast, wind speeds are lower in the valley, particularly over the slopes on the north-western side. A coherent vortex spans
nearly the entire width of the valley, with faster rotation and higher wind speeds during Laseyer gusts, particularly near the
valley floor (colored lines in Fig. 9a,c). Additionally, there is a clear variation of the pressure anomaly with height, relative to
a hydrostatic atmosphere in geostrophic balance (Fig. 9b,d). Over the mountain crests near points A, B, and C, a pronounced
negative pressure anomaly is simulated near the surface, decreasing rapidly in amplitude with altitude. The horizontal vortex
in the valley is linked to a negative pressure anomaly in its center, where its amplitude decreases towards the surface and with
increasing altitude. Meanwhile, over the downstream escarpment, the positive pressure anomaly is strongest near the surface
and also decreases in amplitude with height. In the Laseyer gust composite, the negative pressure anomaly in the valley center
is strongly amplified compared to the simulation mean (colored lines in Fig. 9b,d).

So far, the composites presented the average conditions at the time of Laseyer gusts. However, although the simulation
mean can be considered stationary, the gust composite is transient and has a temporal evolution. Here, we are interested
in this evolution, again averaged over all gust events, during a few minutes before and after Laseyer gusts. Therefore, we
examine time-lagged gust composites during the period $-250\,\mathrm{s}$ to $+250\,\mathrm{s}$ in Fig. 10. Additionally, horizontal sections through
composites at times $-30\,\mathrm{s}$ and $+30\,\mathrm{s}$ can be found in Figs. S1 and S2, respectively. At the location Wasserauen in the valley, the



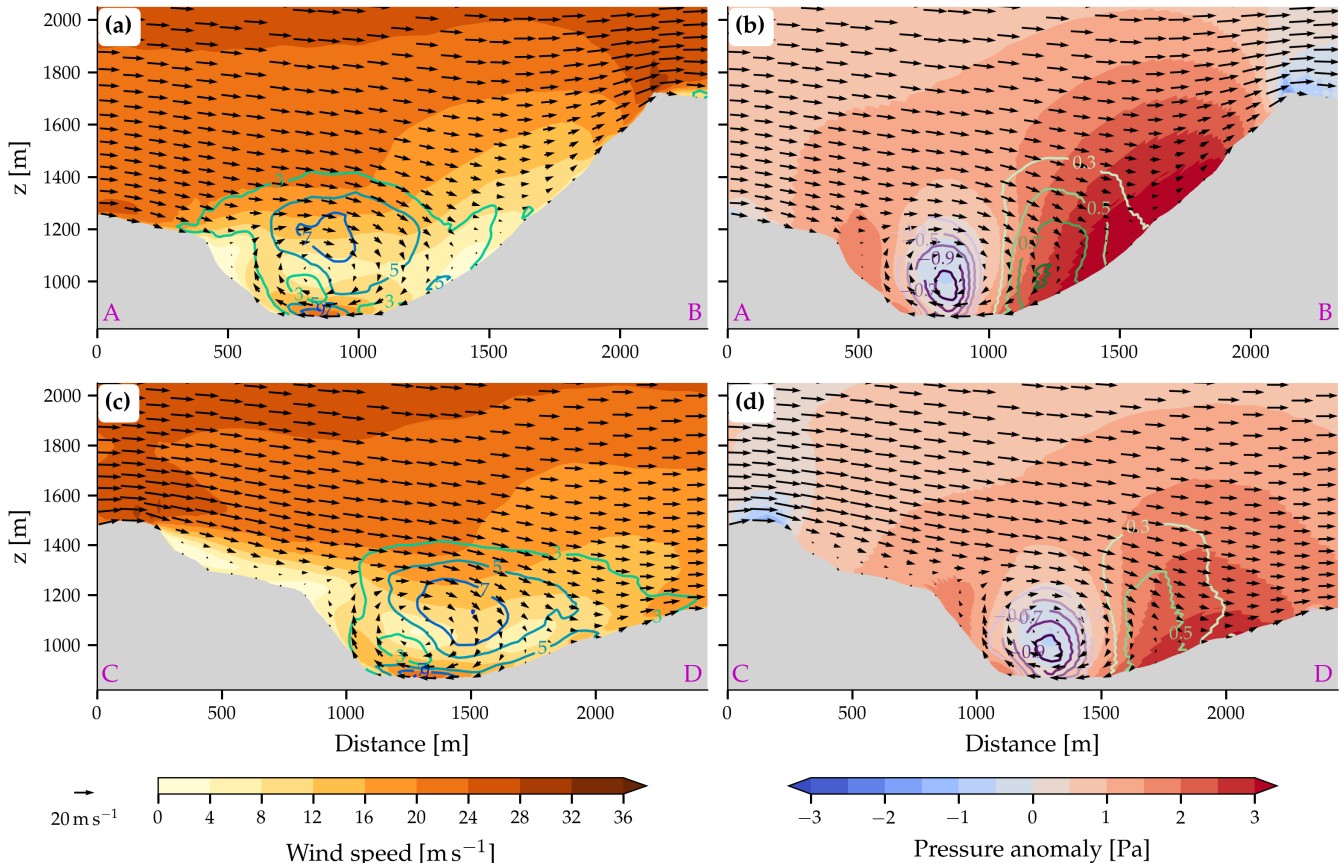

**Figure 9.** Vertical sections across the valley for the Laseyer gust composite. The sections are along the lines **(a,b)** AB and **(c,d)** CD indicated in Fig. 8a. Shown are **(a,c)** three-dimensional wind speed and **(b,d)** pressure anomaly relative to an atmosphere in geostrophic balance over flat terrain. Black arrows denote the wind speed parallel to the plane of the cross-section and the colored lines represent the deviation from the simulation mean of **(a,c)** wind speed (in $\mathrm{m\,s^{-1}}$) and **(b,d)** pressure (in hPa).

surface wind speed exhibits a clear peak at composite time $t_c = 0\,\mathrm{s}$. It rapidly increases before and decreases after the Laseyer

gust (Fig. 10a). The wind speed exceeds the mean of the scalar wind speed for approximately $100\,\mathrm{s}$. Notably, the highest wind speeds originate from a more southerly direction than the mean winds (Fig. 10a). Typically, surface winds blow more from the south before reaching their maximum intensity, while they tend to turn more easterly after the highest wind speeds pass the location Wasserauen.

      Also the winds in the column above Wasserauen demonstrate a clear temporal evolution (Fig. 10b). Between composite times

$t_c \approx -90\,\mathrm{s}$ and $t_c \approx 60\,\mathrm{s}$, the strong north-westerly winds penetrate deeper into the Schwende valley. Furthermore, during this period, the atmosphere at an altitude of approximately $1400\,\mathrm{m}$ exhibits a markedly stronger downward motion than during preceding and subsequent times. This clearly indicates that the depth of the vortex in the valley varies with the surface wind speed. When wind speeds are high at the surface, the vortex is shallower, and when wind speeds are weak at the surface, the

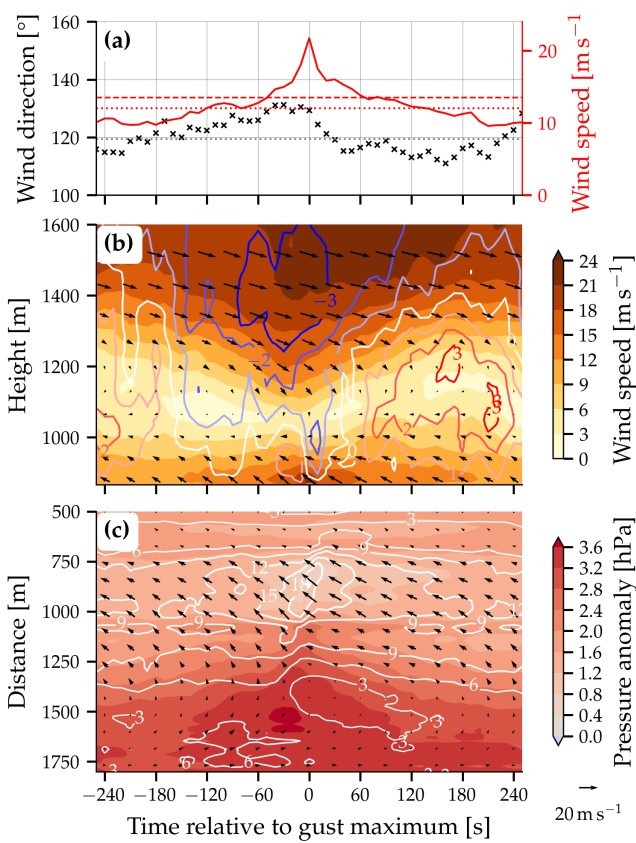

**Figure 10.** Temporal evolution of the Laseyer gusts. Shown are time-lagged composites of **(a)** the surface wind at the location Wasserauen in the valley (black crosses are wind direction, solid red line wind speed), **(b)** the vertical profiles of wind speed (color shaded), horizontal wind (black arrows), and vertical wind (colored lines, in $\mathrm{m\,s^{-1}}$) above the location Wasserauen, and **(c)** the pressure anomaly (color shaded) and surface wind (black arrows is horizontal component, white lines indicate wind speed in $\mathrm{m\,s^{-1}}$) along the central part of the line AB shown in Fig. 8a. The black and red dotted lines in **(a)** illustrate the direction and speed of the mean wind, while the red dashed line represents the mean scalar wind speed. The vertical axis in **(c)** is the same as the horizontal axis in Fig. 9a,b.



vortex is deeper. The Laseyer gust composite's temporal evolution exhibits a distinct pattern of vertical wind near the surface
(Fig. 10b). The air rises ahead of the highest wind speed and descends after the highest wind speeds have passed the location of
Wasserauen. This pattern is consistent with mass conservation as the horizontally convergent (approaching high wind speeds)
and divergent (departing high wind speeds) flow at the surface has to be compensated by vertical motion. Lastly, we analyze the
pressure anomaly (again compared to the atmosphere in geostrophic balance over flat terrain) along the central part of line AB
(indicated in Fig. 8a and Fig. 9a,b), with the results illustrated in Fig. 10c. The highest wind speeds propagate from the south-
east to the north-west over time. The minimum in pressure anomaly coincides with the highest wind speeds, consistent with
the Bernoulli theorem. However, the highest positive pressure occurs about $30\,\mathrm{s}$ earlier on the escarpment to the south-west of
the valley. Closer to the valley center, the largest positive pressure anomaly occurs near the composite time $t_c = 0\,\mathrm{s}$.

Based on our Laseyer gust composite analysis, we hypothesize that the strong reversed surface winds and the strong positive
pressure anomaly on the downstream escarpment are caused by an amplifying interplay and superposition of two mechanisms.
Firstly, the flow separates on the upstream mountain ridge (Fig. 9). This flow separation is facilitated by the salient edge of the
upstream ridge, strong winds, and the nearly neutral stratification of the atmospheric boundary layer. This flow separation is
associated with a recirculation region and an adverse pressure gradient. Secondly, the strong winds impinging on the down-
stream escarpment create a positive pressure anomaly and an associated vortex at low levels. Both of these mechanisms result
in an adverse pressure gradient and a reversed flow at the valley floor. Their superposition then leads to a very strong pressure
gradient and high wind speeds at the valley floor.

To illustrate that the simulated pressure anomaly can cause strong winds at the valley floor, we estimate the wind speed
caused by acceleration related to the pressure gradient force. To keep the calculation simple, we assume a pressure gradient
that remains constant over a few minutes. This yields as the acceleration by the pressure gradient

$$\frac{Du}{Dt} = \frac{\partial u}{\partial t} + u\frac{\partial u}{\partial x} \approx -\frac{1}{\rho}\frac{\partial p}{\partial x}. \tag{1}$$

To integrate Eq. 1 between two points along the air parcel trajectory, $x_0$ and $x_1$, we assume that the air parcel is at rest at its
original location $u(x_0) = 0$, which results in

$$u(x_1) \approx \sqrt{-2\Delta p/\rho}, \tag{2}$$

where $\Delta p = p(x_1) - p(x_0)$. By plugging in the values from the Laseyer gust composite of the pressure difference between two
points in the valley ($\Delta p \approx -3.5\,\mathrm{hPa}$) and a mean density of $\rho \approx 1.16\,\mathrm{kg\,m^{-3}}$, a wind speed of $u(x_1) \approx 24.6\,\mathrm{m\,s^{-1}}$ is obtained.
This wind speed is similar to the wind speed at the valley floor in the gust composite, supporting the claim that the high wind
speeds are caused by the strong pressure gradient perpendicular to the valley axis.

## 3.4 Effect of changing the roughness length

The forest on the downstream escarpment may cause a stronger retardation of the flow and a strengthening of the positive
pressure anomaly. Therefore, it could indirectly affect the gust strength simulated at the valley floor. To test the potential
influence of the increased roughness length of the forests in the valley and on the valley slopes, we conducted two simulations





in which we removed the forest on either the upstream or the downstream mountains. To do so, as described in Sect. 2.2.2, some areas with forest are replaced with the roughness length corresponding to grass. The setup of these simulations is otherwise identical to the reference simulation presented in Sect. 3.2 and 3.3.

In general, the two simulations with altered roughness length fields exhibit a very similar mean surface wind field as the reference simulation (Fig. S3). Small differences are only visible in the regions where the forest is replaced by grass due to the decreased momentum flux at the surface. The maximum gusts at the valley floor in the sensitivity simulations with reduced roughness are slightly weaker than in the reference simulation. However, this difference is relatively small and may be affected by a sampling bias resulting from the discrete model output and limited simulation period. Therefore, we can conclude that altering the roughness length of the two forest patches does not significantly impact the flow in the valley.

## 3.5   Effect of lowering the downstream mountain

As a second sensitivity experiment, we investigate the impact of the height of the downstream mountain. The Laseyer gust composite in the reference simulation leads us to hypothesize that the strong pressure difference and highest wind speeds are partially caused by the presence of the downstream mountain. To test this hypothesis, we perform a simulation with the height of the downstream mountain reduced by 500 m. The simulation uses the topography depicted in Fig. 2c and otherwise identical
initial and lateral boundary conditions as those of the reference simulation.

In the simulation with a lowered downstream mountain, the mean flow still exhibits reversed flow at the valley floor, although it is much weaker compared to the reference simulation (Fig. 11, see also Fig. 5a for comparison). Wind speeds on the valley floor reach values between $20 \, \mathrm{m \, s^{-1}}$ and $30 \, \mathrm{m \, s^{-1}}$ (not shown), which are lower than the peak wind speeds in the reference simulation, which exceed $30 \, \mathrm{m \, s^{-1}}$. Moreover, while Fig. 11b still demonstrates a notable pressure anomaly difference across
the valley, it is approximately 40% weaker than the pressure anomaly difference in the reference simulation (compare with Fig. 9b, noting that the pressure anomaly of the Laseyer gust composite is depicted). Therefore, the height of the downstream mountain is critical for the strong pressure anomaly difference and the high wind speeds at the valley floor. In summary, this additional simulation supports the explanation of the mechanism suggested in Sect. 3.3 and highlights the importance of both the upstream ridge and the downstream mountain in causing and intensifying the reversed flow in the valley.

## 4   Sensitivity to topography smoothing and ambient flow conditions

This section investigates the impact of topography smoothness and changes in ambient flow conditions on the flow in the narrow valley. First, we analyze the sensitivity of the flow to small changes in the topography to assess the robustness of the results with respect to details of the topographic forcing. Second, by changing the ambient flow conditions, we investigate their influence on flow reversal and high wind speeds at the valley floor.



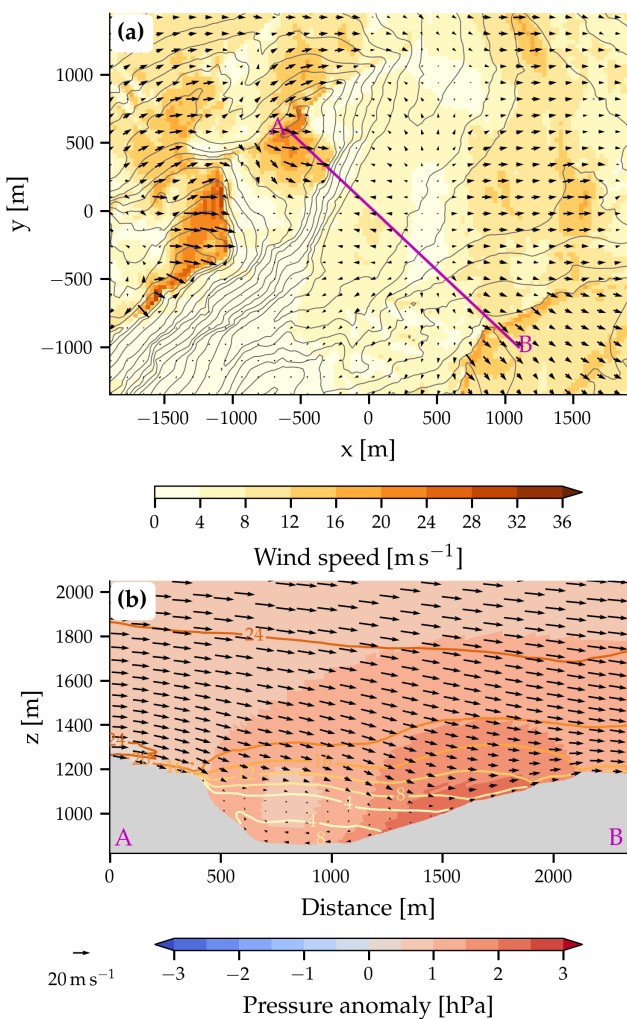

**Figure 11.** Mean flow situation in the simulation with a reduced height of the downstream mountain. Shown are **(a)** surface wind (arrows, color shaded for wind speed) and the height of the topography (gray lines, interval 50 m) and **(b)** the pressure anomaly compared to an atmosphere in geostrophic balance over flat terrain (color shaded), three-dimensional wind speed (colored lines, in m s$^{-1}$), and the wind in the plane of the cross-section (arrows). The location of the vertical cross-section is indicated by the magenta line in **(a)**.



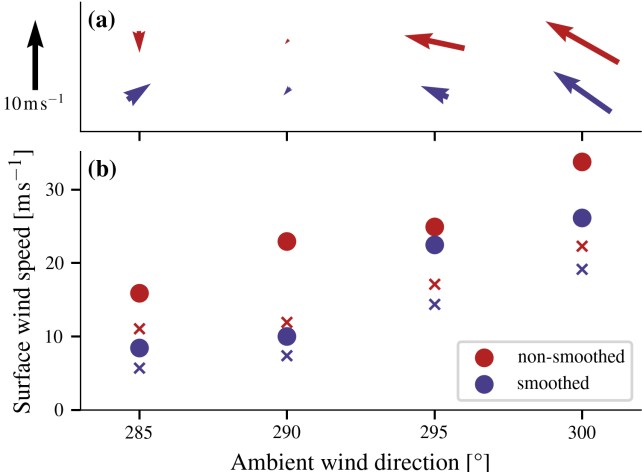

**Figure 12.** Effect of smoothing topography in simulations with an ambient geostrophic wind of $U_g = 20\,\mathrm{m\,s^{-1}}$ from 285°, 290°, 295°, and 300°. Shown are **(a)** mean surface wind and **(b)** the maximum (dots) and 95th percentile (crosses) of the wind speed from an easterly to south-easterly wind direction ($65° \leq \varphi \leq 165°$) at the location of the measurement station Wasserauen in the valley. The black arrow in the top left corner serves as a reference for the lengths of the arrows depicted in **(a)**.

## 4.1 Sensitivity to smoothing the topography

The sensitivity of the flow in the valley to smoothing the topographic field is presented using a few selected simulations that either directly use the interpolated topography field or additionally apply smoothing.

In atmospheric models using terrain-following coordinates, topographic fields are frequently smoothed to remove topographic forcings which originate at scales close to the grid spacing. Smoothing or filtering techniques can improve numerical stability of the model by reducing maximum slopes (e.g., Klemp et al., 2003; Schmidli et al., 2018) or reduce computation time (e.g., Berg et al., 2017, 2018). However, smoothing has also drawbacks such as larger altitude biases of grid points. In a study using simulations performed with the COSMO model at a horizontal grid spacing of $1.1\,\mathrm{km}$, Schmidli et al. (2018) found that reducing topographic filtering led to an improved simulation of valley winds. Liu et al. (2020) found, using the WRF model, that high-resolution topographic data is the most important driving factor for a successful fine-scale wind simulation.

Two series of simulations are presented here, one with and another without topography smoothing. The simulations that use topography smoothing have maximum slopes of up to 70.5°, whereas refraining from topography smoothing results in maximum slopes of up to 77.4° and a more accurate representation of the height of the mountain and ridge tops. The sensitivity investigation was conducted in four different simulations with an ambient geostrophic wind speed of $U_g = 20\,\mathrm{m\,s^{-1}}$ from 285°, 290°, 295°, and 300°. These simulations were selected due to their strong sensitivity to ambient flow conditions (see Sect. 4.3 and Fig. 14). Consequently, they are likely to be sensitive to changes in the representation of the topography.

Significant differences arise between simulations with smoothed and non-smoothed topography under the same ambient wind conditions (Fig. 12). Despite the overall similarity in mean surface wind in the valley for all four ambient wind directions,





some notable variations occur. For instance, with an ambient wind direction of 285°, the simulation with smoothed topography exhibits a weak south-westerly mean wind, while the non-smoothed simulation exhibits a weak northerly mean wind. For more

north-westerly ambient wind directions of 295° and 300°, the simulations model south-easterly winds in the valley. However, the mean wind is slightly weaker and more southerly in the simulations with smoothed topography. Furthermore, the maximum gusts and 95th percentile of gusts, both from an easterly to south-easterly wind direction, exhibit a clearer difference between the two sets of simulations. The strongest south-easterly winds at the valley floor are weaker when the topography is smoothed.

These findings are consistent with the effect of the smoothing procedure in LES, as demonstrated by Berg et al. (2017) and

Berg et al. (2018). Berg et al. (2017) found that the specific smoothing procedure had a large impact on the size and position of a recirculation zone. Additionally, differences in wind between a simulation with smoothed topography and one with steeper and more realistic terrain were revealed (Berg et al., 2018).

### 4.2  Sensitivity to ambient wind conditions

To investigate how the flow in the valley reacts to changes in the ambient wind conditions, we perform a series of LES with

varying initial and lateral boundary conditions. These simulations were performed using the smoothed topography as we only later realized the excellent quality of the simulations without topography smoothing. We conducted 16 simulations with the ambient wind speeds ranging from $12\,\mathrm{m\,s^{-1}}$ to $24\,\mathrm{m\,s^{-1}}$ and wind directions ranging from 285° to 330°. These wind speeds and directions are chosen based on observations of wind speeds in the lower free atmosphere during events of the Laseyer (see Sprenger et al., 2018). A wind speed of $24\,\mathrm{m\,s^{-1}}$ is used as an example of an ambient wind speed that rarely occurs but could

demonstrate the situation under extreme conditions. To simplify the comparison of different simulations, we only compare a few aggregated characteristics of the flow in the valley. A more detailed comparison of the surface wind fields in the 16 simulations can be found in Fig. S4 in the supplementary material.

The wind roses in Fig. 13 illustrate, for all simulations, surface wind at the center of the valley and roughly in the middle of the region where the strongest reversed winds occur. The flow in the valley is steady and from a narrow south-westerly

directional window for a westerly ambient wind direction of 285° and moderate winds (Fig. 13m,n). As the ambient wind speed increases, wind direction becomes more variable. At extreme ambient wind speeds of $24\,\mathrm{m\,s^{-1}}$, weak winds may come from a north-westerly or northerly direction at the valley floor (Fig. 13p). However, for the majority of the time, the flow in the valley remains south-westerly. A slight 15° shift in ambient wind direction to 300° results in a different flow response in the narrow valley. For this ambient wind direction, winds in the valley come from the south-east and can be quite strong for

strong to extreme ambient wind speeds (Fig. 13k,l). An even more northerly ambient wind has little influence on the flow in the valley and leads to stronger south-easterly winds even for weaker ambient wind conditions. With an ambient wind direction of 330°, the winds in the valley blow from a south-easterly direction ($U_g = 12\,\mathrm{m\,s^{-1}}$, Fig. 13a) or from a north-easterly to easterly direction ($U_g = 16\,\mathrm{m\,s^{-1}}$ to $24\,\mathrm{m\,s^{-1}}$, Fig. 13b-d). Whereas increasing the ambient wind speed generally results in stronger winds at the valley floor, the variability of the wind direction remains similar for different ambient wind speeds.

To investigate the transition from south-westerly to south-easterly winds in the valley in more detail, we conducted two additional simulations. The mean surface wind in the simulations with an ambient geostrophic wind speed of $U_g = 20\,\mathrm{m\,s^{-1}}$



**Figure 13.** Wind roses at the location of the measurement station Wasserauen on the valley floor in the 16 simulations with different ambient wind conditions. The columns illustrate the results for the different ambient wind speeds of 12, 16, 20, and 24 m s$^{-1}$, and the rows correspond to the ambient wind direction of 330°, 315°, 300°, and 285°. The radial axes give the frequency (gray number is %) of the wind direction in each wind direction bin of 10° width. Note the different radial axis in **(i,m,n,p)**.

**Figure 14.** Sensitivity of the mean surface wind to different ambient wind directions. Shown are simulations with an ambient wind speed of $20\,\mathrm{m\,s^{-1}}$ and an ambient wind direction of **(a)** 285°, **(b)** 290°, **(c)** 295°, and **(d)** 300°. The colored lines illustrate the difference compared to the mean of the four simulations (magnitude of wind difference vector of 3, 6, 9, and $12\,\mathrm{m\,s^{-1}}$) and the gray lines denote the height of the topography (interval is $50\,\mathrm{m}$).





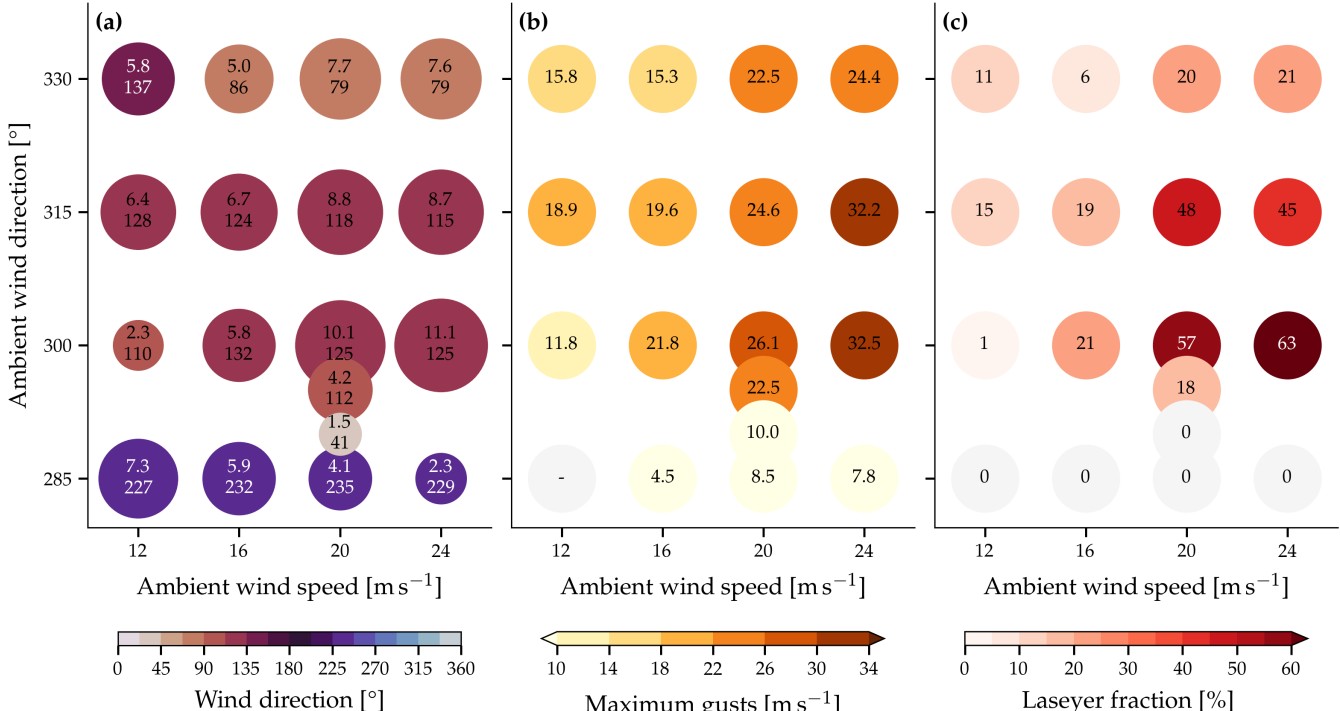

**Figure 15.** Sensitivity of the flow at the location of the measurement station Wasserauen in the valley to changing ambient wind conditions. Shown are **(a)** speed (size of dots and number in first row, in $\mathrm{m\,s^{-1}}$) and direction (color and number in second row, in degrees) of mean wind, **(b)** maximum gusts in the Laseyer window ($65° \leq \varphi \leq 165°$, values in $\mathrm{m\,s^{-1}}$), and **(c)** Laseyer fraction (in %, see text for definition).

from 285°, 290°, 295°, and 300° is depicted in Fig. 14. When the ambient wind direction is westerly at 285°, the flow in the valley is generally south-westerly and down-valley (Fig. 14a). The flow in the lowest part of the valley is weak and turns westward on the escarpment to the south-east of the valley. For a more north-westerly ambient wind direction of 290°, the flow

in the lowest part of the valley is almost stagnant on average (Fig. 14b), whereas in the upper part of the valley, weak down-valley flow occurs. While for an ambient wind direction of 285°, the flow is predominantly south-westerly, the wind direction in the valley exhibits larger variation for an ambient geostrophic wind from 290° (not shown). The wind is mostly weak and from a northerly to easterly direction, but it can also have a southerly or westerly component. For an ambient geostrophic wind from 295°, the surface wind in the valley is north-easterly to south-easterly. The mean surface wind speed at the valley floor is

considerably weaker than for an ambient wind direction of 300°. Nonetheless, moderate to strong wind speeds from an easterly to south-easterly direction can occur (see also Fig. 15). For an ambient wind direction of 295°, the wind at the valley floor has a westerly component for a small fraction of the output times (not shown). This is in contrast to simulation with an ambient wind direction of 300°, where the winds are exclusively easterly to south-easterly (see also Fig. 13k).

   Important additional aggregated values for the simulations with changing ambient wind conditions are presented in Fig. 15.

The mean wind direction at a point location in the valley (Fig. 15a) clearly transitions from the south-westerly regime at an





ambient wind direction of 285° to the south-easterly regime for more north-westerly ambient wind directions. It is noteworthy that the mean wind speed in the valley decreases with increasing ambient wind speed in the south-westerly regime, while it increases with increasing ambient wind speed in the south-easterly regime. We hypothesize that, as the wind speed increases, the flow in the valley shifts slightly towards the flow regime occurring for more north-westerly ambient wind conditions. Both

the mean wind speed and maximum gusts occurring during the simulation increase with increasing ambient wind speed (Fig. 15b). Notably, both the mean wind speed and maximum gusts generally decrease for more northerly ambient winds within the south-easterly regime in the valley. Finally, Fig. 15c depicts the Laseyer fraction at the point location in the valley. The Laseyer fraction is defined as the proportion of time steps when the wind direction $\varphi$ is easterly to south-easterly ($65° \leq \varphi \leq 165°$) and the wind speed is at least $10\,\mathrm{m\,s^{-1}}$. Similar to the mean and maximum wind speeds, an increase in ambient wind speeds

generally results in an increase in the Laseyer fraction. The Laseyer fraction is generally highest for an ambient wind direction of 300° and decreases for more northerly ambient winds.

Our sensitivity analysis provides results consistent with previous findings based on measurements from the valley and upstream ridge (see Sect. 1 and Sprenger et al., 2018). The occurrence of the local windstorm is highly dependent on strong north-westerly ambient winds and is sensitive to changes in the ambient wind direction. Westerly winds with a southerly

component or a only weak northerly component are not conducive to the occurrence of the reversed flow at the valley floor. Although, in this study, we do not directly compare our semi-idealized LES to observations, these results with the striking sensitivity to the ambient flow increase confidence in our results because they are reasonable if compared with previous knowledge gained from observations.

### 4.3 Sensitivity to the stability

We here investigate two additional simulations with an ambient geostrophic wind speed of $U_g = 20\,\mathrm{m\,s^{-1}}$ from 300° with increased and decreased atmospheric stability (see also Sect. 2.2.1). Two general characteristics of the simulated flow change in these simulations. Firstly, the turbulent boundary layer is shallower with increased stability and deeper with decreased stability (not shown). Secondly, the amplitude and vertical wavelength of the gravity wave triggered by the entire Alpstein massif change. When stability increases, the amplitude of the gravity wave increases while the vertical wavelength decreases

(not shown). Conversely, in the simulation with decreased stability, the amplitude decreases while the vertical wavelength increases.

In the simulation with increased stability ($N = 1.5 \cdot 10^{-2}\,\mathrm{s^{-1}}$), the mean flow at the valley floor is calm and does not reverse compared to the flow further aloft (not shown). At the location closest to the measurement station Wasserauen, the flow is mostly easterly, but westerly and northerly flow also occurs during the simulation. In contrast, the simulation with

decreased atmospheric stability ($N = 7 \cdot 10^{-3}\,\mathrm{s^{-1}}$) exhibits a surface wind field that is much more similar to the simulation with $N = 10^{-2}\,\mathrm{s^{-1}}$. Significant differences between these three simulations are apparent in the mean flow situation in a vertical section across the valley axis (Fig. 16). At higher elevations, the impact of the increased amplitude of the gravity wave is evident in the higher wind speeds occurring in the simulation with increased stability (Fig. 16a) compared to the simulation with $N = 10^{-2}\,\mathrm{s^{-1}}$ (Fig. 16b) and particularly the simulation with $N = 7 \cdot 10^{-3}\,\mathrm{s^{-1}}$ (Fig. 16c). Furthermore, the location of



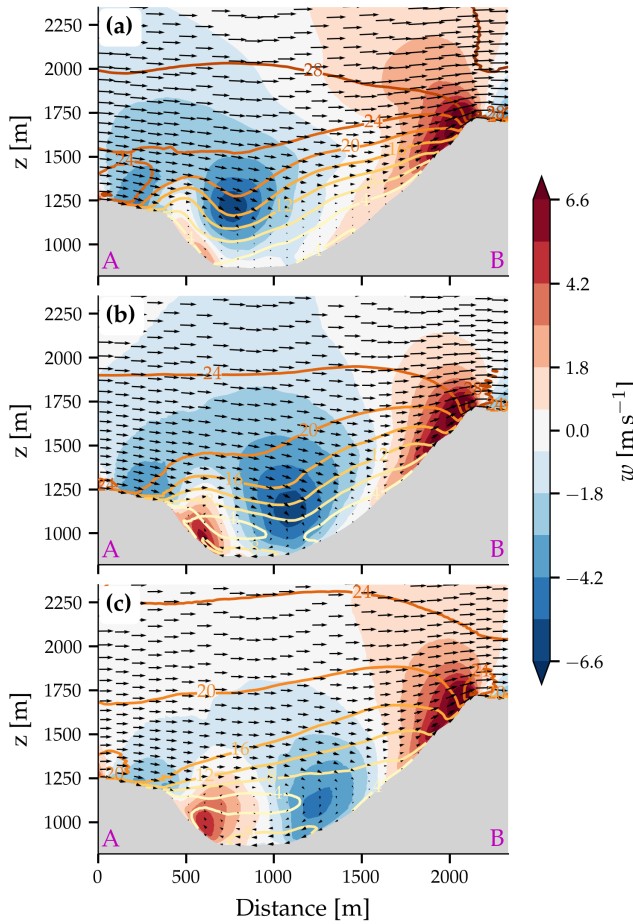

**Figure 16.** Mean flow in the vertical cross-section across the valley. Shown are vertical velocity $w$ (color shaded), the wind along the plane of the cross-section (black arrows), and the magnitude of the wind vector (colored lines, in $\mathrm{m\,s^{-1}}$) for the simulations with a stability of **(a)** $N = 1.5 \cdot 10^{-2}\,\mathrm{s^{-1}}$, **(b)** $N = 10^{-2}\,\mathrm{s^{-1}}$, and **(c)** $N = 7 \cdot 10^{-3}\,\mathrm{s^{-1}}$. The location of the vertical section is depicted in Fig. 8a.

the descending motion in the valley shifts with changing stability. For higher atmospheric stability, the descending motion occurs nearer to the ridge upstream (Fig. 16a), whereas in the simulation with low stability, it shifts towards the downstream escarpment (Fig. 16c). As a result, in the simulations with $N = 10^{-2}\,\mathrm{s^{-1}}$ and $N = 7 \cdot 10^{-3}\,\mathrm{s^{-1}}$, the recirculation region in the valley extends over (almost) the entire width of the valley, while it remains confined to the north-western part of the valley in the simulation with $N = 1.5 \cdot 10^{-2}\,\mathrm{s^{-1}}$. Thus, in the most stable simulation, the recirculation region induced by flow separation

on the upstream ridge and the vortex caused by the impinging flow on the downstream ridge do not reinforce each other. This is particularly evident in the simulation with $N = 7 \cdot 10^{-3}\,\mathrm{s^{-1}}$, where the two effects induced by the upstream and downstream mountain align well, and despite the lower wind speed above the valley, the recirculation flow in the valley remains strong.



Therefore, these three LES suggest that static stability and in turn the location and size of the recirculation regions, as well as their interplay, are crucial for the occurrence and strength of the reversed flow at the valley floor.

The shift in the location of the downward motion in our simulations is generally consistent with theory, with an increase in stability leading to smaller and shallower recirculation regions (Hunt and Snyder, 1980; Baines, 2022). However, it should be noted that in all three simulations, the boundary layer is well mixed and has an almost neutral stratification (not shown). We argue that the stability of the free atmosphere above the turbulent boundary layer has a similar effect as the stability of the boundary layer, and the increased stability results in a compression of the recirculation region.

## 5   Summary and conclusions

Semi-idealized LES are performed over the Alpstein massif in north-eastern Switzerland to investigate the mechanism of the Laseyer and its sensitivity to changes in the ambient flow conditions. The LES successfully simulate the flow reversal in the valley observed during strong north-westerly winds and provide insight into the mechanism of the flow reversal and the high wind speeds.

Composites for times with the highest wind speeds at the valley floor from a south-easterly direction reveal that the very strong gusts are caused by a large horizontal pressure difference perpendicular to the valley axis ($\Delta p \approx 3.5 \, \mathrm{hPa}$ over a distance of about $1 \, \mathrm{km}$). Grønås and Sandvik (1999) found nonhydrostatic pressure anomalies of similar magnitude in simulations of flow over complex terrain with comparable horizontal and vertical scales of the valley and the surrounding mountains. However, these simulations were conducted at a considerably lower spatial resolution and only demonstrated steady flow solutions (Grønås and Sandvik, 1999). In other valleys with similar flows, the wind exhibits remarkable variability over $10 \, \mathrm{min}$ (Grønås and Sandvik, 1999). With our LES at a horizontal grid spacing of $\Delta = 30 \, \mathrm{m}$, we are able to capture (part of) the temporal variability and explicitly simulate the strong gusts. Additionally, the gust composites depict a coherent vortex in the narrow valley with its axis parallel to the valley axis. We explain this strong vortex by the amplifying interplay of a recirculation region caused by flow separation on the upstream mountain and a recirculation region caused by the impinging flow on the downstream escarpment. This explanation is supported by additional simulations with either modified ambient static stability or with a lowered height of the downstream mountain. A LES with increased atmospheric stability, resulting in a lower extent of the atmospheric boundary layer, exhibits, on average, only weak winds at the valley floor. In this simulation with increased stability, the downward motion induced by the recirculation region behind the upstream mountain is located in the center of the valley and, thus, does not amplify with the recirculation region caused by the impinging flow on the downstream escarpment. In the simulation with a lowered height of the downstream mountain, flow reversal still occurs, but with reduced amplitude.

In the second part of our study, we performed a series of LES to investigate the sensitivity of the flow in the narrow valley to the ambient flow conditions. Our LES indicate that stronger ambient geostrophic winds generally result in higher maximum wind speeds at the valley floor. The direction of the ambient wind is even more critical, and we only simulate strong reversed surface winds at the valley floor for a small range of wind directions. A slight $15°$ turning of the ambient winds results in a distinctly different flow response in the narrow valley. Specifically, when the ambient wind direction is $285°$, the flow at





the valley floor is south-westerly. For an ambient wind direction of 300°, however, strong south-easterly winds are simulated. Additionally, the stability of the atmosphere is a crucial factor in the occurrence of flow reversal and high wind speeds at the valley floor. In our simulations, a weaker static stability led to stronger and particularly deeper reversed flow at the valley floor.

Together, the set of simulations with modified topography, static stability, and flow characteristics, emphasize the significance of ambient flow conditions on the occurrence and strength of flow reversal, as well as the complicated three-dimensional interaction of the flow in the narrow valley and its surroundings. A similarly complicated interplay of flow structures may also be important for other dynamically-driven flows in steep and complex terrain.

Finally, sensitivity experiments were performed about the impact of smoothing the topography on the flow response in the narrow valley. Although the flow in the valley is generally similar for simulations with a smoothed topography, the strength of

the reversed flow, and particularly the highest wind speeds, are reduced. Schmidli et al. (2018) demonstrated the importance of filtering the topography in high-resolution mesoscale simulation, where higher-resolution topographic data (i.e., weaker filtering) improved performance. Similarly, Berg et al. (2017) illustrated that the specific smoothing procedure has a large impact on the size and position of a recirculation zone in LES at $20\,\mathrm{m}$ grid spacing. Furthermore, Berg et al. (2018) demonstrated that the wind differed between a LES with smoothed topography and one with steep and realistic terrain. Therefore, our findings

are qualitatively in line with previous research on the role of topographic details and smoothing, however, in our case the effect of smoothed topography does not change the interpretation of the mechanism of the Laseyer.

Our LES in complex terrain demonstrate the importance of flow separation and recirculation for the mean flow in a narrow valley. Additionally, we simulate the large temporal variability of the in-valley flow and the transient nature of the very high wind speeds in or close to the recirculation region. Maximum wind speeds exceeding $35\,\mathrm{m\,s^{-1}}$ were simulated, which could

cause damage to infrastructure. This emphasizes the importance of examining flow separation and recirculation regions not only as mean flow structures but also to investigate their temporal variability and, in particular, the magnitude of the highest wind speeds.

In summary, our semi-idealized LES setup allows us to investigate the mechanism of the local windstorm and its sensitivity to the ambient flow conditions in a controlled environment. Several results from our semi-idealized LES compare qualitatively

well with the known climatology of the local windstorm. However, the idealized atmospheric state also limits the applicability of the simulated results to the real atmosphere and prevents a direct comparison with observations. We therefore discuss here some of the main limitations. The ambient atmospheric conditions in our LES remain constant without any temporal, horizontal, or vertical variation. Small-scale variations are common in the real atmosphere, and particularly the vertical variation of the wind speed or direction could have an influence on the flow in the valley. Also, Laseyer winds occur in specific synoptic

situations that are transient and therefore can lead to rapidly changing ambient conditions. Furthermore, our LES did not include any moist processes. This is particularly relevant as the climatology of Laseyer conditions indicates that precipitation is frequently observed during Laseyer events (Sprenger et al., 2018). Since our dry simulations qualitatively capture many of the observed features of the Laseyer, we dare to hypothesize that mainly dry dynamics govern the occurrence of the flow reversal and high wind speeds at the valley floor and that adding diabatic effects would only marginally affect the main results.

In simulations of a similar phenomenon by Grønås and Sandvik (1999), however, the inclusion of wet processes increased



the strength of the reversed flow. Moreover, simulations in idealized settings have demonstrated that heating or cooling at the surface can affect the size and strength of a recirculation region (Dörnbrack and Schumann, 1993; Allen and Brown, 2006). Therefore, we hypothesize that the mechanism of the local windstorm remains unchanged with the inclusion of moist processes, but these processes might affect the strength of the gusts. To address these limitations, LES of observed Laseyer case studies

with time-dependent boundary conditions and cloud physics would be necessary – which is planned in our next project.

Despite these limitations, our semi-idealized LES and analysis provide valuable findings. Our results show that PMAP can handle accurately and in a robust manner the complex and steep terrain with slopes of up to 77.4°. Such steep slopes occur at the considered decameter resolution when the original topography data in the simulation domain is not filtered or smoothed. Using these model capabilities, our results provide valuable and unprecedented insights into the dynamical mechanism of the

Laseyer and its sensitivity to the ambient flow conditions. The performed LES clearly point to the key role of the detailed topographic and surface forcing for estimating the potential damage and risk of locally severe winds. The results also show the need for measurements that can capture the high temporal variability and three-dimensional nature of the flow in the narrow valley.

## Appendix A: Vertical level and topography specifications

**A1   Vertical levels**

For the specification of the terrain-following coordinate surfaces, we use an algorithm following Klemp (2011). However, the smoothing parameters are specified differently than in Klemp (2011). First, we do not set a minimum allowed fractional vertical grid spacing but set a constant minimum allowed vertical grid spacing $\Delta z = 15\,\mathrm{m}$ equal to the vertical grid spacing close to the surface. Furthermore, the parameter $\beta_k$ is set according to

$$\beta_k = 0.8 \min\left( \frac{\zeta_k}{4h_m}, 1 \right),$$   (A1)

and $M_k = 25$ for our application. Here, $\zeta_k$ is the constant vertical height of the level $k$, $h_m$ is the maximum terrain height, and $M_k$ is the number of iterations. Lastly, as an attenuation profile $A(\zeta)$ for the terrain influence, we use

$$A(\zeta) = \begin{cases} \cos^2\left( \frac{\pi}{2} \frac{\zeta}{z_H} \right), & \text{for } \zeta < z_H, \\ 0, & \text{for } \zeta \geq z_H, \end{cases}$$   (A2)

where $z_H = 12\,\mathrm{km}$ is the height where the grid transitions to constant height coordinate surfaces.

**A2   Smoothing**

To get the smoothed topography for the simulations presented in Sect. 4, the following procedure was applied. Initially, the topography of the elevation model was interpolated to a grid with $\Delta = 45\,\mathrm{m}$ (1.5 times coarser than grid used for the simulations). The topography based on this coarser grid was then smoothed using a 8[th] order filter following Shapiro (1975) and 40





passes to this filter. This operation damps all waves with wavelengths smaller than about $4\Delta$ on this coarser grid and retains the
amplitude of longer waves. This smoothed topography on the coarser grid was then interpolated to the grid for the simulations.
The transition from the real topography in the inner part of the domain to the constant height of the topography at the domain
boundaries was applied (see Appendix A3) and then the smoothing using a 8th order Shapiro filter and 40 passes to the filter
was repeated. As a final step, 100 passes to a 10th order Shapiro filter were performed. The overall effect of the smoothing
procedure is to damp waves with wavelengths smaller than about $6\Delta$, while the amplitude of longer waves is almost completely
retained.

## A3 Transition at lateral boundaries

At the domain boundaries, the topography is set to a constant value of $h_b = 600\,\text{m}$, such that the uniform boundary conditions
can easily be prescribed. The transition from the constant height of the topography $h_b$ at the boundaries to the real topography
$h_r$ in the interior of the domain was performed according to

$$h_t = h_b f_t + h_r (1 - f_t),\tag{A3a}$$

$$f_t = \begin{cases} \frac{1}{2}\left[1 + \cos\left(\frac{i\pi}{n_t}\right)\right], & \text{if } i \leq n_t, \\ 0, & \text{otherwise,} \end{cases}\tag{A3b}$$

where $h_t$ is the height of the topography used in the simulations, $i$ counts the grid points from the boundaries, and $n_t = 50$ is
the width in grid points of the transitional region.

## A4 Changed topography of the downstream mountain

For the simulation to investigate the role of the downstream mountain, we adapted the topography in the target region. The
topography was changed according to

$$h_a = h_c + f_b(h_t - h_c),\tag{A4a}$$

$$f_b = 1 - \frac{5}{8}\begin{cases} \cos\left(\pi r/2\right), & \text{if } r \leq 1, \\ 0, & \text{otherwise,} \end{cases}\tag{A4b}$$

$$r = \max\left(\left|\frac{(x - x_0)\cos\alpha + (y - y_0)\sin\alpha}{a}\right|,\right.$$

$$\left.\left|\frac{(x - x_0)\sin\alpha - (y - y_0)\cos\alpha}{b}\right|\right),\tag{A4c}$$

where $h_a$ is the adapted topography, $h_t$ is the original interpolated topography, $h_c = 850\,\text{m}$ is a baseline height, $x_0 = 2\,751\,700\,\text{m}$
and $y_0 = 1\,238\,400\,\text{m}$ give the central location of the bell $f_b$ in Swiss reference coordinates, $a = 2100\,\text{m}$ and $b = 1400\,\text{m}$ give
the half-widths of the bell, and $\alpha = 55°$ is the angle by which the bell is rotated.



*Author contributions.* NK performed the simulations and the data analysis and led the writing of the manuscript. CK, MS, and HW super-
vised the work and reviewed and edited the manuscript. CK helped with the setup of the simulations and the needed model extensions. MS
and HW aided with the interpretation of results.

*Competing interests.* At least one of the (co-)authors is a member of the editorial board of *Weather and Climate Dynamics*. The authors have
no other competing interests to declare.

*Acknowledgements.* This work was supported by the Platform for Advanced Scientific Computing (PASC) project "KILOS" (Kilometer-scale
nonhydrostatic global weather forecasting with IFS-FVM) and the ETH Board. All simulations were carried out at ETH Zurich's scientific
and high-performance cluster EULER. We thank Stefano Ubbiali for his valuable help in installing and running the model on the EULER
cluster. AI tools, specifically DeepL Write and ChatGPT, were utilized for language editing and stylistic adjustments in the preparation of
this paper.



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
