# Peer review of "Revealing the dynamics of a local Alpine windstorm using large-eddy simulations"

_EGUsphere, 2024_

## Referee Comment (RC2)

**Review of "Revealing the dynamics of a local Alpine windstorm using large-eddy simulations" by Krieger et al.**

In the article the authors show large-eddy simulations for a local high-impact wind storm in a narrow valley in Switzerland using a new numerical model. The simulations are performed in a semi-idealized framework investigating the influence of the ambient wind field and air mass stratification as well as topographic influences. The manuscript is very well-written and structured, has high relevance and fits the scope of WCD. I only have a few comments and questions on the manuscript listed below.

Personally I am very excited about this new model and especially the capacity of dealing with steep slope angles.

**Specific comments**

- Sec. 1: The introduction is very interesting and extensive, but could benefit from some streamlining. I suggest that especially the state-of-knowledge part between L39 and L96 could be more compact and streamlined towards the knowledge gap that is addressed in the study.

- Sec. 2.1: I assume there will be more information once Kühnlein et al. is published, in the meantime, however, more general information on the model is appropriate. Is the model mainly designed for LES applications or can it also be used in meso-scale applications? Can the model be run in a real-case setting as well? In this study moist processes and radiation are neglected, but does PMAP in general include microphysics and radiation schemes? Can you elaborate a bit more, why PMAP is able to handle such steep slope angles with terrain-following coordinates compared to other models? Is there an upper limit to the slope angles in PMAP? Apart from that, I want to encourage the authors to provide the model publicly available in line with the policies for data and software by WCD.

- L 166: You use a uniform profile of stratification and wind as initial conditions. However, in the case of Kyrill the upstream air mass shows a clear multi-layer structure with a higher-stability layer above 700 hPa (see Sprenger et al. 2018, Fig 4) as well as vertical shear in wind speed and direction (see Fig 1). How representative are the simulations presented here for the Kyrill-case given the strong influence of vertical variations in Scorer-parameter on gravity wave propagation?

- L285-290: I liked the spectral analysis and think it would deserve a more extensive evaluation, especially as it is stated, that the pulsating gusts are a key finding that was not known before. In the spectral analysis some of my questions remained open: for the layers below $\sim$100 m in Fig. 7 the spectral density between $\sim 4 \cdot 10^{-3}$ and $\sim 10^{-2}\,\mathrm{s}^{-1}$ shows a local minimum which is not present for the higher levels. Do you have an explanation for this? Does that correspond to the thickness of the Laseyer-gust layer? As you show later that the gusts likely are linked to a enhanced downward motion and higher wind speeds in the valley atmosphere at about crest height (Fig. 9), do you see there a similar maximum in spectral density for a pulsation period of $\sim$450 s as well? I would like to see more on the mechanism behind the pulsating gusts especially as they have a period similar to reports in earlier studies (e.g. Peltier and Scinocca 1990; Belušić et al. 2007; Tollinger et al. 2019), but seem not to be caused by Kelvin-Helmholtz instabilities or gravity wave breaking as proposed in these studies.

- L358-365: I think in the conclusion of the governing mechanisms you should also include the role of the enhanced horizontal and vertical winds in the valley atmosphere at crest height mentioned in L346.

- Sec. 4.1: It might make sense to shift this section after Sec. 4.2 as you are referring to different ambient flow directions which have not been discussed yet.

- L448: unspecific location description with "... roughly in the middle ..." , please specify more and check throughout the manuscript for these unclear location names. Consider also adding markers for important locations like Wasserauen station in the plots.

- L450-453: How much can we trust the results shown here, given the northerly mean wind for non-smoothed terrain at 285° ambient wind shown in Sec. 4.1, Fig. 12? How does the wind rose or the surface wind field for the non-smoothed case look like? Apart from that: could you elaborate more on the penetration mechanism in the 285° case? Is the recirculation/rotor still present in this case or is the air penetrating the valley somewhere further up-valley?

[Figure]

Figure 1: Radiosounding from Stuttgart on January 19, 2007 12 UTC. Taken from University of Wyoming (https://weather.uwyo.edu/upperair/sounding.html)

- L478-479: I do not understand this sentence: How can the flow in the valley resemble more the flow regime in north-westerly ambient cases, when the flow in the valley shows more northerly components but no south-easterly?

- L505: can you show the surface wind fields for the experiments in the supplement?

- L580: as mentioned earlier, the uniform profiles in wind and stratification should be added as a potential limitation here.

**References**

Belušić, D., M. Žagar, and B. Grisogono, 2007: Numerical simulation of pulsations in the bora wind. *Quarterly Journal of the Royal Meteorological Society*, **133 (627)**, 1371–1388, doi:10.1002/qj.129, URL https://rmets.onlinelibrary.wiley.com/doi/10.1002/qj.129.

Peltier, W. R. and J. F. Scinocca, 1990: The Origin of Severe Downslope Windstorm Pulsations. *Journal of the Atmospheric Sciences*, **47 (24)**, 2853–2870, doi:

10.1175/1520-0469(1990)047⟨2853:TOOSDW⟩2.0.CO;2, URL `http://journals.ametsoc.org/doi/10.1175/1520-0469(1990)047<2853:TOOSDW>2.0.CO;2`.

Tollinger, M., A. Gohm, and M. O. Jonassen, 2019: Unravelling the March 1972 northwest Greenland windstorm with high-resolution numerical simulations. *Quarterly Journal of the Royal Meteorological Society*, **145 (725)**, 3409–3431, doi:10.1002/qj.3627, URL `https://rmets.onlinelibrary.wiley.com/doi/10.1002/qj.3627`.

---

## Author Comment (AC1)

**Final author comments for paper egusphere-2024-3461**

**Revealing the dynamics of a local Alpine windstorm using large-eddy simulations**

by Nicolai Krieger, Heini Wernli, Michael Sprenger, and Christian Kühnlein

January 23, 2025

We thank both reviewers for their constructive comments, which helped us to further improve the clarity of our study. In the following, we carefully address all comments of the referees. They are shown in **blue** and our replies in **black**. Mentioned references are listed at the end of the document.
* * *
**Reviewer 1**

**Summary**

This paper investigates the dynamics of the Laseyer, a potentially damaging local windstorm occurring in a deep, narrow valley in northeastern Switzerland, using semi-idealized, dry Large-Eddy Simulations. Previous observations from measurement stations at the upstream ridge and in the valley resulted in a 10-year climatology, suggesting that the extreme wind gusts in the valley are because of flow separation and a flow reversal during strong ambient flow. Indeed, the simulations validate this hypothesis by successfully reproducing the quasi-periodic flow reversal along with extreme wind gusts. Through the careful selection of Laseyer-characteristic episodes for composite analysis, the study reveals the spatio-temporal structure and the key mechanism driving this potentially damaging storm: a mutually reinforcing interaction between a recirculation region caused by lee-side flow separation at the upstream ridge, and a vortex induced by a positive pressure anomaly from the impinging flow at the downstream ridge. Furthermore, through sensitivity analyses of these mechanisms with respect to changes of ambient flow properties and topographical features, the authors demonstrate that the intensity of local windstorms, such as the Laseyer, is critically influenced by atmospheric conditions and that reproducing the highest wind speeds requires simulations with detailed representations of the topography.

**General comments**

This work is a unique and valuable contribution to understanding terrain-induced circulations in complex orography, particularly their temporal variability and potential to damaging wind gusts. The manuscript is well-written, clearly structured, and supported by carefully prepared high-quality

plots. The methods are thoroughly described, ensuring reproducibility. Despite the comprehensive analysis, the authors effectively highlight the key findings. Given the high quality of the manuscript, I only have a few minor comments and suggestions for further refinement.

**Reply**: Thank you for your positive evaluation of our manuscript. Your insightful comments and constructive suggestions were very helpful, and we appreciate the time and effort you devoted to this review. We are confident that your feedback has helped us strengthen the manuscript and we hope that the revised version will align with your expectations.

**Specific and technical comments**

**Reviewer Comment 1.1** — Line 53: Add Prestel and Wirth (2016) to the listed references. They showed similar results for idealized simulations with a three-dimensional mountain.

**Reply 1.1**: Thank you for the suggestion. We will add the reference.

**Reviewer Comment 1.2** — Line 57: Add Prestel and Wirth (2016) to the listed references. They showed that for rather shallow mountains without surface friction, lee-side flow separation was only possible with strong stratification. In contrast, with neutral stratification, lee-side flow separation was only possible with surface friction.

**Reply 1.2**: Thank you for the suggestion. We will add the reference.

**Reviewer Comment 1.3** — Line 120: In addition to the influence of ambient flow conditions, the description of the sensitivity analysis is missing the influence of topographical conditions including vegetation (roughness length), mountain height and smoothing of the topography.

**Reply 1.3**: Thank you so much for your constructive comment. We will include the influence of topographical conditions in the description of the sensitivity analysis both in L120 and L124.

**Reviewer Comment 1.4** — Line 140: The year of publication is missing for "Kühnlein et al.".

**Reply 1.4**: Thank you for this observation. As the manuscript Kühnlein et al. is currently only in preparation, it does not yet have a year. In line with WCDs citation guidelines, we will either reference a preprint (if available when we will hand in our revised manuscript) or entirely remove this citation.

**Reviewer Comment 1.5** — Line 172: I am wondering how variations of the potential temperature perturbations, e.g. its vertical depth or its decay with height, might affect the simulation outcomes. What criteria were used to parameterize these perturbations? Were they based on observational data in this region? Additionally, without employing these perturbations to accelerate the development of turbulence, how much longer would the spin-up period have needed to reach the statistical steady state you deemed suitable for your analysis?

**Reply 1.5**: The application of the potential temperature perturbations primarily aims to accelerate the transition from laminar flow at the lateral boundaries to resolved turbulence towards the domain interior. This approach follows the established method described in Muñoz-Esparza et al. (2014) and Muñoz-Esparza et al. (2015), which has been validated in various subsequent studies (e.g., Jähn et al.,

2016; Jiang et al., 2017; Muñoz-Esparza et al., 2017; Muñoz-Esparza and Kosović, 2018; Lee et al., 2019; Connolly et al., 2021; Hawbecker and Churchfield, 2021). By leveraging this proven technique, we enhance the quality of the resolved turbulence and, consequently, improve the overall reliability and quality of our simulation. Importantly, this method targets turbulence development and does not affect the time required to attain a statistically steady state.

The amplitude of the perturbations was set based on the optimal scaling identified in Muñoz-Esparza et al. (2015). However, compared to them, we set the same random perturbation over the entire depth of the perturbation region. Although complex terrain accelerates the transition to resolved turbulence, a turbulence generation method can still significantly shorten the fetch (Connolly et al., 2021; Hawbecker and Churchfield, 2021), reinforcing our decision to employ a turbulence generation method.

It is challenging to anticipate the effects of modifying the parameters of the potential temperature perturbations on the simulation outcomes. Altering the vertical depth or the decay of the perturbation amplitude with height would likely directly influence the turbulence intensity and, consequently, also the (turbulent) structure of the (upper part of the) boundary layer (see also e.g., Kumar et al., 2024). For instance, a deeper perturbation layer or no decay with height would likely result in more vigorous turbulence at a height of $500\,\mathrm{m}$ above ground. However, as the applied method was used in various previous publications (see citations further above), we found no compelling incentive to investigate the role of the perturbations. Moreover, given that the cell perturbation method (Muñoz-Esparza et al., 2014; 2015) was designed to only affect the resolved turbulence and not the mean flow, we speculate that there would be only weak sensitivity to parameters (within their reasonable range) specifying the potential temperature perturbations.

We expect that the perturbations applied at the initial time within the domain interior have a much smaller effect. Although these perturbations may have reduced the time needed to reach a statistical steady state, we believe that the spin-up period of one hour would have been enough even in the absence of these perturbations. Particularly, as in all simulations, within one hour the air is advected at least once through the entire domain (using a wind speed of $12\,\mathrm{m\,s^{-1}}$ and the longest diagonal through the domain, which spans approximately $39\,\mathrm{km}$).

To clarify the role of the potential temperature perturbations, their expected effect, and particularly the role of the perturbations during the simulation, we will revise the two paragraphs between L172 and L182.

**Reviewer Comment 1.6** — Line 268: Can you clarify what is meant by the term "streamlined component"? As I understand it, this would correspond to the wind direction specified in line 209 (300 degrees). However, the time series near the measurement station at Wasserauen, as shown in Figure 6, indicates that this wind direction rarely occurs. Are you instead referring to the wind direction sector mentioned?

**Reply 1.6**: The time series in Fig. 6 was rotated to give both streamwise (along-wind) and spanwise (perpendicular to the mean wind direction) wind components. These components were identified based on the local mean wind direction, which is in this case south-easterly.

**Reviewer Comment 1.7** — Line 469 & 470: Add "(Fig. 14c)" and "(Fig. 14 d)" at the end of both sentences.

**Reply 1.7**: Thank you for the suggestion. We will add references to both sentences.

**Reviewer Comment 1.8** — Line 475 ff: Change unspecific "at a point location in the valley" to more specific "at the location of the measurement station Wasserauen in the valley".

**Reply 1.8**: Thank you for pointing this out. We will rephrase accordingly.
* * *
**Reviewer 2**

**Review of "Revealing the dynamics of a local Alpine windstorm using large-eddy simulations" by Krieger et al.**

In the article the authors show large-eddy simulations for a local high-impact wind storm in a narrow valley in Switzerland using a new numerical model. The simulations are performed in a semi-idealized framework investigating the influence of the ambient wind field and air mass stratification as well as topographic influences. The manuscript is very well-written and structured, has high relevance and fits the scope of WCD. I only have a few comments and questions on the manuscript listed below. Personally I am very excited about this new model and especially the capacity of dealing with steep slope angles.

**Reply**: Thank you for your positive evaluation and kind feedback on our manuscript. We are thrilled to hear your enthusiasm for our model and its ability to address steep slope angles. Your excitement about this aspect of our work is encouraging and reinforces our belief in the potential impact of the proposed approach. Your thoughtful feedback has been invaluable in refining our manuscript. We sincerely hope that the revised version aligns with your expectations and continues to inspire interest in this topic.

**Specific comments**

**Reviewer Comment 2.1** — Sec. 1: The introduction is very interesting and extensive, but could benefit from some streamlining. I suggest that especially the state-of-knowledge part between L39 and L96 could be more compact and streamlined towards the knowledge gap that is addressed in the study.

**Reply 2.1**: Thank you for your thoughtful suggestion. We appreciate your positive feedback on its interesting scope and agree that the introduction in the original submission was extensive. In response to your comment, we will revise and condense the state-of-knowledge section between L39 and L96.

**Reviewer Comment 2.2** — Sec. 2.1: I assume there will be more information once Kühnlein et al. is published, in the meantime, however, more general information on the model is appropriate. Is the model mainly designed for LES applications or can it also be used in meso-scale applications? Can the model be run in a real-case setting as well? In this study moist processes and radiation are neglected, but does PMAP in general include microphysics and radiation schemes? Can you elaborate a bit more, why PMAP is able to handle such steep slope angles with terrain-following coordinates compared to other models? Is there an upper limit to the slope angles in PMAP?

Apart from that, I want to encourage the authors to provide the model publicly available in line with the policies for data and software by WCD.

**Reply 2.2**:   We appreciate this comment and the reviewer's interest in PMAP. Yes, more information on PMAP for LES will be provided in the upcoming overview publication and the code will then be made publicly available. Independent of this upcoming publication, we believe the model description in Section 2 of the present paper, together with the relevant references about the methods it uses, provide the reader sufficient information to gain a proper understanding of the model formulation.

The present paper comprehensively discusses the dynamics of the Laseyer using the described semi-idealized LES configuration. We hope it is acceptable that a discussion of PMAP's general and future capabilities is beyond the scope of the present paper. Nevertheless, in response to the reviewer's suggestion, we will extend selected parts of the model description in Section 2. Additionally, we will ensure that the information in Section 5 about planned real-case LES of the Laseyer, which is intended to be reported in a follow-up publication, is clearly highlighted.

We further provide here short answers to the reviewer's questions: (i) The model can also be used for meso-scale applications. Even though PMAP-LES is developed with the primary focus on weather prediction using hectometer and finer grid spacing, it can also be applied at coarser resolutions. (ii) Full moist-precipitating processes are already available in PMAP but were not considered in this study; however, the follow-up publication mentioned in Section 5 will include moist processes. The incorporation of a radiation scheme and land model is in preparation but not yet available at the time of writing. Once these parameterizations have been implemented, the model will be able, in principle, to perform realistic regional weather forecasts. (iii) Unfortunately, there is no simple answer why the model can handle steep slopes well. We listed some of the important aspects in the model description part of Section 2, such as the rigorous curvilinear coordinate formulation with a height-based vertical coordinate and the specification of boundary conditions consistently throughout the entire model. Another aspect we believe plays a key role for the stability and accuracy of the model is the particular implicit time integration in all three spatial dimensions on the co-located grid. We have not yet experienced stability issues with extreme slopes, which is encouraging. However, we also note that our experience with extremely steep slopes is limited and we still need to explore many regimes in real configurations at full model complexity.

**Reviewer Comment 2.3** — L 166: You use a uniform profile of stratification and wind as initial conditions. However, in the case of Kyrill the upstream air mass shows a clear multi-layer structure with a higher-stability layer above 700 hPa (see Sprenger et al. 2018, Fig 4) as well as vertical shear in wind speed and direction (see Fig 1). How representative are the simulations presented here for the Kyrill-case given the strong influence of vertical variations in Scorer-parameter on gravity wave propagation?

**Reply 2.3**:   Thank you for your comment and for providing the additional information on the conditions during Kyrill. While the Kyrill case and the associated derailment was used to motivate our investigation of the Laseyer windstorm, our objective was not to replicate the exact conditions during Kyrill. Instead, we specified simple, more general initial conditions that we believe to be conducive to Laseyer conditions. In particular, the uniform stratification closely resembles the mean tropospheric profile during Laseyer conditions (Fig. R1). However, we also note that the vertical structure of the atmosphere during Laseyer conditions can vary significantly (not shown). As our results demonstrate, the uniform profiles are sufficient to produce Laseyer conditions. While a higher-stability layer, like the one during Kyrill, may influence gravity wave propagation, the conditions in the boundary layer and in the lower-most

[Figure]

**Figure R1:** Mean atmospheric profiles of temperature $T$ (black line), dew point temperature (gray), and Brunt-Väisälä frequency $N$ (red) during Laseyer conditions. The profile is extracted from ERA5 data at the location closest to the measurement station Wasserauen. Laseyer conditions were identified between 1998 and 2021 based on conditions of westerly to north-westerly winds at Ebenalp (on the upstream ridge) and easterly to south-easterly winds and gusts of at least $20\,\mathrm{m\,s^{-1}}$ at Wasserauen.

free troposphere, and consequently also the intensity of the windstorm, we believe the fundamental mechanisms driving the windstorm to be consistent and well represented by our simulations.

We will add a clarification in the manuscript regarding our choice of uniform wind and stratification profiles.

**Reviewer Comment 2.4** — L285-290: I liked the spectral analysis and think it would deserve a more extensive evaluation, especially as it is stated, that the pulsating gusts are a key finding that was not known before. In the spectral analysis some of my questions remained open: for the layers below $\sim 100\,\mathrm{m}$ in Fig. 7 the spectral density between $\sim 4\cdot 10^{-3}$ and $\sim 10^{-2}\,\mathrm{s^{-1}}$ shows a local minimum which is not present for the higher levels. Do you have an explanation for this? Does that correspond to the thickness of the Laseyer-gust layer? As you show later that the gusts likely are linked to a enhanced downward motion and higher wind speeds in the valley atmosphere at about crest height (Fig. 9), do you see there a similar maximum in spectral density for a pulsation period of $\sim 450\,\mathrm{s}$ as well? I would like to see more on the mechanism behind the pulsating gusts especially as they have a period similar to reports in earlier studies (e.g. Peltier and Scinocca, 1990; Belušić et al., 2007; Tollinger et al., 2019), but seem not to be caused by Kelvin-Helmholtz instabilities or

[Figure]

**Figure R2:** Spectra of the streamwise velocity component at different heights above the valley floor. The dashed lines represent the power spectral density of the streamwise component of the surface wind ($z = 7.5\,\mathrm{m}$) and wind at $354\,\mathrm{m}$ above ground using a longer segment length of $n_s = 360$. Note that the $y$-axis represents the product of frequency and the power spectral density. The thin gray lines indicate the $-2/3$ spectral slope.

gravity wave breaking as proposed in these studies.

**Reply 2.4**: Thank you for your detailed and thoughtful comment. We do not currently have an explanation for the local minimum in spectral density observed at low levels, which disappears at higher levels. This is an interesting finding that warrants further investigation in future studies.

To address your suggestion, we will include a spectral analysis at crest height (see Fig. R2) to evaluate whether similar quasi-periodic oscillations are present at this level. The results indicate that oscillations of comparable period occur at crest height, supporting the link between these oscillations and the enhanced downward motion and higher wind speeds in the valley atmosphere. We will incorporate this analysis into the manuscript by replacing Fig. 7 with the updated figure (Fig. R2) and discussing the findings.

We agree that a more thorough analysis of the quasi-periodic pulsations, including their mechanisms, could provide valuable insights. However, an investigation into the causes of these pulsations is beyond the scope of the present study. As the pulsations are a novel finding and their causes remain unclear, we will better outline the need for future research on this topic in the manuscript.

**Reviewer Comment 2.5** — L358-365: I think in the conclusion of the governing mechanisms you should also include the role of the enhanced horizontal and vertical winds in the valley atmosphere at crest height mentioned in L346.

**Reply 2.5**: Thank you for this suggestion. We agree and will add the role of the enhanced horizontal and vertical winds at crest height.

**Reviewer Comment 2.6** — Sec. 4.1: It might make sense to shift this section after Sec. 4.2 as you are referring to different ambient flow directions which have not been discussed yet.

**Reply 2.6**: Thank you for this thoughtful suggestion. We understand the potential benefit of introducing the different ambient flow directions earlier. However, we have carefully considered the structure and, while no arrangement is perfect, we believe the current order works well. Particularly, we believe that introducing the simulations with smoothed topography already in Section 4.1 is important, as they are essential for presenting the subsequent results. To address potential confusion, we will revise the manuscript to better clarify its structure.

**Reviewer Comment 2.7** — L448: unspecific location description with "... roughly in the middle ..." , please specify more and check throughout the manuscript for these unclear location names. Consider also adding markers for important locations like Wasserauen station in the plots.

**Reply 2.7**: Thank you for pointing out the lack of specificity. We will replace vague phrases with more precise location references.

**Reviewer Comment 2.8** — L450-453: How much can we trust the results shown here, given the northerly mean wind for non-smoothed terrain at 285° ambient wind shown in Sec. 4.1, Fig. 12? How does the wind rose or the surface wind field for the non-smoothed case look like? Apart from that: could you elaborate more on the penetration mechanism in the 285° case? Is the recirculation/rotor still present in this case or is the air penetrating the valley somewhere further up-valley?

**Reply 2.8**: Thank you for raising these insightful and thought-provoking questions. Your comment has prompted us to further investigate these aspects. The mean surface winds at the valley floor in the simulations forced by an ambient wind direction of 285° are associated with major uncertainty (as you pointed out and already mentioned in Sect. 4.1 and illustrated by Fig. 12). Of the presented simulations, those with an ambient wind direction of 285° show the largest uncertainty in the surface wind, as also illustrated by the wind roses in Fig. R3. However, the differences in wind direction are most pronounced near the surface, while at higher elevations, wind direction in both simulations with non-smoothed and smoothed topography is more consistent (Fig. R4). It is certainly worthwhile to better reflect the similarities and differences between the simulations using smoothed and non-smoothed topography and consequently the uncertainty associated with the mean surface wind in these simulations. To this aim, we will add additional information to the manuscript when presenting these simulations.

Regarding the penetration mechanism: as illustrated by Fig. R4, there is hardly any motion across the valley axis and the winds blow mostly along the valley axis with some downward motion at higher elevations. The main penetration of the air into the valley occurs further up-valley (not shown).

**Reviewer Comment 2.9** — L478-479: I do not understand this sentence: How can the flow in the valley resemble more the flow regime in north-westerly ambient cases, when the flow in the valley shows more northerly components but no south-easterly?

**Reply 2.9**: Thank you for pointing out the potential lack of clarity in the mentioned sentence. While revisiting the manuscript, we agree that this sentence may indeed be confusing and does not contribute to the understanding, as it pertains to a really minor detail. To improve the manuscript's focus and clarity, we will remove this sentence in the revised version.

[Figure]

**Figure R3:** Wind roses at the location of the measurement station Wasserauen on the valley floor in 8 simulations with different ambient wind directions and topography representation. The columns illustrate the results for **(a,c,e,g)** smoothed topography and **(b,d,f,h)** non-smoothed topography, and the rows correspond to the ambient wind direction of 300°, 295°, 290°, and 285°. The radial axes give the frequency (gray number is %) of the wind direction in each wind direction bin of 10° width.

[Figure]

**Figure R4:** Mean flow in vertical cross-sections across the valley in the simulation forced by a geostrophic wind of $20\,\mathrm{m\,s^{-1}}$ from 285° with **(a,c)** smoothed topography and **(b,d)** non-smoothed topography. The sections are along the lines **(a,b)** AB and **(c,d)** CD indicated in Fig. 8a. Shown are vertical velocity $w$ (color shaded), the wind along the plane of the cross-section (black arrows), and the magnitude of the wind vector (colored lines, in $\mathrm{m\,s^{-1}}$).

**Reviewer Comment 2.10** — L505: can you show the surface wind fields for the experiments in the supplement?

**Reply 2.10**: Thank you for this suggestion. We will add Fig. R5 to the supplement.

**Reviewer Comment 2.11** — L580: as mentioned earlier, the uniform profiles in wind and stratification should be added as a potential limitation here.

**Reply 2.11**: Thank you for this suggestion. We will mention the uniform stratification as a potential limitation. The missing vertical variations in wind speed and / or wind direction were already mentioned in L 581.

[Figure]

**Figure R5:** Sensitivity of the mean surface wind to changes in the static stability. Presented are simulations with an ambient wind speed of $20\,\mathrm{m\,s^{-1}}$ from $300°$ with a stability of **(a)** $N = 1.5 \cdot 10^{-2}\,\mathrm{s^{-1}}$, **(b)** $N = 1.0 \cdot 10^{-2}\,\mathrm{s^{-1}}$, and **(c)** $N = 0.7 \cdot 10^{-2}\,\mathrm{s^{-1}}$.

**References**

Belušić, D., M. Žagar, and B. Grisogono, 2007: Numerical simulation of pulsations in the bora wind. *Quarterly Journal of the Royal Meteorological Society*, **133 (627)**, 1371–1388, DOI: `10.1002/qj.129`.

Connolly, A., L. van Veen, J. Neher, B. J. Geurts, J. Mirocha, and F. K. Chow, 2021: Efficacy of the cell perturbation method in large-eddy simulations of boundary layer flow over complex terrain. *Atmosphere*, **12 (1)**, 55, DOI: `10.3390/atmos12010055`.

Hawbecker, P. and M. Churchfield, 2021: Evaluating terrain as a turbulence generation method. *Energies*, **14 (21)**, 6858, DOI: `10.3390/en14216858`.

Jähn, M., D. Muñoz-Esparza, F. Chouza, O. Reitebuch, O. Knoth, M. Haarig, and A. Ansmann, 2016: Investigations of boundary layer structure, cloud characteristics and vertical mixing of aerosols at Barbados with large eddy simulations. *Atmospheric Chemistry and Physics*, **16 (2)**, 651–674, DOI: `10.5194/acp-16-651-2016`.

Jiang, P., Z. Wen, W. Sha, and G. Chen, 2017: Interaction between turbulent flow and sea breeze front over urban-like coast in large-eddy simulation. *Journal of Geophysical Research: Atmospheres*, **122 (10)**, 5298–5315, DOI: `10.1002/2016JD026247`.

Kumar, M., A. Jonko, W. Lassman, J. D. Mirocha, B. Kosović, and T. Banerjee, 2024: Impact of momentum perturbation on convective boundary layer turbulence. *Journal of Advances in Modeling Earth Systems*, **16 (2)**, e2023MS003643, DOI: `10.1029/2023MS003643`.

Lee, G.-J., D. Muñoz-Esparza, C. Yi, and H. J. Choe, 2019: Application of the cell perturbation method to large-eddy simulations of a real urban area. *Journal of Applied Meteorology and Climatology*, **58 (5)**, 1125–1139, DOI: `10.1175/JAMC-D-18-0185.1`.

Muñoz-Esparza, D. and B. Kosović, 2018: Generation of inflow turbulence in large-eddy simulations of nonneutral atmospheric boundary layers with the cell perturbation method. *Monthly Weather Review*, **146 (6)**, 1889–1909, DOI: 10.1175/MWR-D-18-0077.1.

Muñoz-Esparza, D., B. Kosović, J. van Beeck, and J. Mirocha, 2015: A stochastic perturbation method to generate inflow turbulence in large-eddy simulation models: Application to neutrally stratified atmospheric boundary layers. *Physics of Fluids*, **27 (3)**, 035102, DOI: 10.1063/1.4913572.

Muñoz-Esparza, D., B. Kosović, J. Mirocha, and J. van Beeck, 2014: Bridging the transition from mesoscale to microscale turbulence in numerical weather prediction models. *Boundary-Layer Meteorology*, **153 (3)**, 409–440, DOI: 10.1007/s10546-014-9956-9.

Muñoz-Esparza, D., J. K. Lundquist, J. A. Sauer, B. Kosović, and R. R. Linn, 2017: Coupled mesoscale-LES modeling of a diurnal cycle during the CWEX-13 field campaign: From weather to boundary-layer eddies. *Journal of Advances in Modeling Earth Systems*, **9 (3)**, 1572–1594, DOI: 10.1002/2017MS000960.

Peltier, W. R. and J. F. Scinocca, 1990: The origin of severe downslope windstorm pulsations. *Journal of the Atmospheric Sciences*, **47 (24)**, 2853–2870, DOI: 10.1175/1520-0469(1990)047<2853:TOOSDW>2.0.CO;2.

Tollinger, M., A. Gohm, and M. O. Jonassen, 2019: Unravelling the March 1972 northwest Greenland windstorm with high-resolution numerical simulations. *Quarterly Journal of the Royal Meteorological Society*, **145 (725)**, 3409–3431, DOI: 10.1002/qj.3627.